# Achieving Personalized Federated Learning with Sparse Local Models

## Abstract

Federated learning (FL) is vulnerable to heterogeneously distributed data, since a common global model in FL may not adapt to the heterogeneous data distribution of each user. To counter this issue, personalized FL (PFL) was proposed to produce dedicated local models for each individual user. However, PFL is far from its maturity, because existing PFL solutions either demonstrate unsatisfactory generalization towards different model architectures or cost enormous extra computation and memory. In this work, we propose federated learning with personalized sparse mask (FedSpa), a novel PFL scheme that employs personalized sparse masks to customize sparse local models on the edge. Instead of training an intact (or dense) PFL model, FedSpa only maintains a fixed number of active parameters throughout training (aka sparse-to-sparse training), which enables users' models to achieve personalization with cheap communication, computation, and memory cost. We theoretically show that with the rise of data heterogeneity, setting a higher sparsity of FedSpa may potentially result in a better convergence on its personalized models, which also coincides with our empirical observations. Comprehensive experiments demonstrate that FedSpa significantly saves communication and computation costs, while simultaneously achieves higher model accuracy and faster convergence speed against several state-of-the-art PFL methods.

## 1 Introduction

Data privacy raises increasingly intensive concerns, and governments have enacted legislation to regulate the privacy intrusion behavior of mobile users, e.g., the General Data Protection Regulation (Voigt & Von dem Bussche, 2017). Traditional distributed learning approaches, requiring massive users' data to be collected and transmitted to a central server for training model, soon may no longer be realistic under the increasingly stringent regulations on users' private data. On this ground, federated learning (FL), a distributed training paradigm emerges as a successful solution to cope with privacy concerns, which allows multiple clients to perform model training within the local device without the necessity to exchange the data to other entities. In this way, the data privacy leakage problem could be potentially relieved.

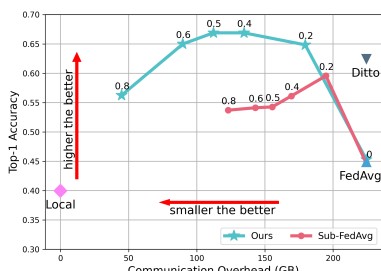

Figure 1: Performance of FedSpa and several baselines w.r.t. communication cost in Non-IID setting. Numbers above FedSpa and Sub-FedAvg are sparsity.

Despite the promising prospect, several notorious issues are afflicting practical performance of FL:

- *The global model produced by weight average (or FedAvg and its non-personalized variants) exhibits unsatisfactory performance in a Non-IID data distribution setting.* To alleviate this problem, the most popular idea is to integrate personalized features into the global model, and produce dedicated model for each local distribution. However, how to make this integration is an open problem that remains unresolved. Prior works on personalized FL (PFL) zero in this issue, but the existing methods either demonstrate weak generalization towards different model architectures (Arivazhagan et al., 2019), or require extra computation and storage (Li et al., 2021).
- *The communication and training overhead is prohibitively high for both the FL and PFL.* Clients in FL/PFL responsible for model training are mostly edge-devices with limited computation capacity and low bandwidth, and may not be powerful enough to fulfill a modern machine learning

task with large deep neural networks. Existing studies (Li et al., 2020; Vahidian et al., 2021) integrate model compression into FL/PFL to save communication and computation overhead. However, both methods embrace the technique of dense-to-sparse training, which still requires a large amount of communication at the beginning of training. In addition, how to effectively aggregate the dynamic sparse models is another challenging problem that remains unresolved.

In this work, we propose FedSpa (see Figure 2), which has two key features to counter the above two challenges: (**i**) FedSpa does not deploy a single global model, but allows each client to own its unique sparse model masked by a personalized mask, which successfully alleviates the Non-IID challenge. (**ii**) FedSpa allows each client to train over an evolutionary sparse model with constant sparsity[1] throughout the whole federated training process, which consistently alleviates the computation overhead of clients. Besides, all the local models in FedSpa are sparse models, which requires a smaller amount of communication cost in each communication round. Theoretically, we conclude that with the rise of Non-IID extent, setting a higher sparsity may result in a better convergence on the personalized models of FedSpa. Empirically, in the Non-IID setting, we demonstrate that FedSpa **accelerates** the convergence (respectively 76.2% and 38.1% less communication rounds to reach the best accuracy of FedAvg (McMahan et al., 2016) and Ditto (Li et al., 2021)), **in-**

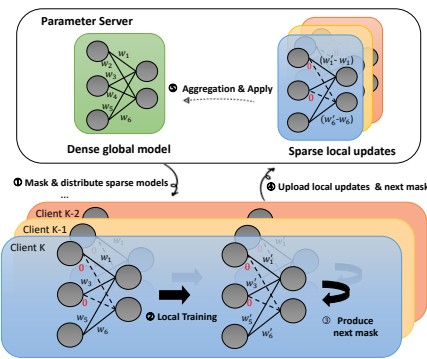

Figure 2: Overview of FedSpa. Firstly, the server masks and distributes the sparse weights. Secondly, clients do local training on a constantly sparse model. Thirdly, clients upload the sparse updates and being aggregated by the server.

**creases** the final accuracy (up to 21.9% and 4.4% higher accuracy than FedAvg and Ditto, respectively), **reduces** the communication overhead (50% less parameters communicated than the dense solutions), and **lowers** the computation (15.3% lower floating-point operations (FLOPs) than algorithms trained with fully dense model). To the end, we summarize our contribution as:

- We present a novel formulation of the sparse personalized FL (SPFL) problem, which can be applied to various network architectures by enforcing personalized sparse masks to a global model.

- We propose a solution dubbed as FedSpa to solve the SPFL problem. By our novel design, FedSpa reduces the communication and computation overhead of the general FL solution.

- Two sparse-to-sparse mask searching techniques are integrated as plugins of our solution. To adapt our PFL training context, we modify the DST-based mask searching technique to enable a warm-start of the searching process, which achieves superior performance.

- We theoretically present the convergence analysis of the personalized models. Experimental results demonstrate the superiority of FedSpa and also coincides with the theoretical conclusion – with the rise of data heterogeneity, setting a higher sparsity of FedSpa may potentially result in a better convergence on its personalized models.

## 2 RELATED WORKS

Federated learning (FL) (McMahan et al., 2016) is seriously afflicted by the issue of heterogeneously distributed (or Non-IID) data. Personalized FL (PFL), initiated by recent literature (Li et al., 2021; Arivazhagan et al., 2019), is shown to be effective to counter this issue of FL. In this work, we propose an alternative yet effective way to enhance PFL with personalized sparse models.

### 2.1 PERSONALIZED FEDERATED LEARNING

We categorize PFL into five genres. Firstly, PFL via layer partition, e.g., FedPer (Arivazhagan et al., 2019), LG-FedAvg (Liang et al., 2020), FedRep (Collins et al., 2021), is to divide the global model layers into shared layers and personalized layers. For the shared layers, weights average as in FedAvg is adopted, while for personalized layers, models are trained only locally and will not

---

[1]Sparsity specifies the ratio of parameters that are set to 0 (or inactive) in a model.

be exchanged with others. Secondly, PFL via regularization, e.g., Ditto (Li et al., 2021), L2GD (Hanzely & Richtárik, 2020) is to add a proximal term on the local model to force the local model and global model closely in the local model fine-tuning stage. Similarly, Sarcheshmehpour et al. (2021); SarcheshmehPour et al. (2021) propose a total variation (TV) regularization to form the network lasso (nLasso) problem, and primal-dual methods adapted from (Jung, 2020) are proposed to solve the nLasso problems. Thirdly, PFL via model interpolation, e.g., MAPPER (Mansour et al., 2020), APFL (Deng et al., 2020) achieves personalization by linearly interpolating the weights of the cluster (global) model and local model as the personalized model. Fourthly, PFL via transfer learning, e.g., FedMD (Li & Wang, 2019), FedSteg (Yang et al., 2020), and Fedhealth (Chen et al., 2020), is to either use model and domain-specific local fine-tuning or knowledge distillation to adapt the global model into the personalized model. Finally, PFL via model compression, e.g., LotteryFL (Li et al., 2020) and Sub-FedAvg (Vahidian et al., 2021), achieves personalization via employing principle model compression techniques, such as weight pruning and channel pruning, over the shared global model.

## 2.2 SPARSE DEEP NEURAL NETWORKS

Methods to Sparsify neural networks can be classified into two genres: dense-to-sparse methods and sparse-to-sparse methods. Dense-to-sparse methods train from a dense model, and compress the model along the training process. Iterative pruning, first proposed by Frankle & Carbin (2018), shows promising performance in dynamically searching for a sparse yet accurate network. Recently, sparse-to-sparse methods have been proposed to pursue training efficiency. Among them, dynamic sparse training (DST) (Bellec et al., 2018; Evci et al., 2020; Liu et al., 2021) is the most successful technique that allows sparse networks, trained from scratch, to match the performance of their dense equivalents. Stemming from the first work – sparse evolutionary training (Mocanu et al., 2018; Liu et al., 2020), DST has evolved as a class of sparse training methods absorbing many advanced techniques, e.g., weight redistribution (Mostafa & Wang, 2019; Dettmers & Zettlemoyer, 2019), gradient-based regrowth (Dettmers & Zettlemoyer, 2019; Evci et al., 2020), and extra weight exploration (Jayakumar et al., 2020; Liu et al., 2021).

Our work also achieves personalization via model compression. We emphasize that three main progresses are made towards SOTA compression-based PFL: (**i**) We rigorously formulate the sparse personalized FL problem, filling the gap left by the prior works. (**ii**) While prior works either vaguely describe their model aggregation as "aggregating the Lottery Ticket Network via FedAvg" (Li et al., 2020), or "taking the average on the intersection of unpruned parameters in the network" (Vahidian et al., 2021), we explicitly formulate the aggregation as averaging the sparse update from clients. (**iii**) Both the two prominent prior works use the idea of iterative pruning to prune the network from dense to sparse. We instead provide two sparse-to-sparse training alternatives to plug in our solution, which largely reduces the costs of communication at the beginning of the training process, and exhibits remarkable performance.

## 3 PROBLEM FORMULATION

We assume a total number of $K$ clients within our FL system, and we consistently use $k$ to index a specific client. First, we give a preliminary introduction on the general FL and PFL problem.

**General FL problem.** Let $\boldsymbol{w} \in \mathbb{R}^d$ be the global weight. General FL takes the formulation as below

$$(\text{P1}) \qquad \min_{\boldsymbol{w}} \tilde{f}(\boldsymbol{w}) = \frac{1}{K} \sum_{k=1}^{K} \left\{ \tilde{F}_k(\boldsymbol{w}) := \mathbb{E}[\mathcal{L}_{(\boldsymbol{x},y) \sim \mathcal{D}_k}(\boldsymbol{w}; (\boldsymbol{x}, y))] \right\},$$

where $\mathcal{D} = \mathcal{D}_1 \cup \cdots \cup \mathcal{D}_K$ is the joint distribution of $k$ local heterogeneous distributions, data $(\boldsymbol{x}, y)$ is uniformly sampled according to distribution $\mathcal{D}_k$ wrt. client $k$, and $\mathcal{L}(\cdot; \cdot)$ is the loss function.

**Ultimate PFL Problem.** Let $\{\boldsymbol{w}_k\}$ be the personalized models. The ultimate PFL is defined as:

$$(\text{P2}) \qquad \min_{\{\boldsymbol{w}_1, \cdots, \boldsymbol{w}_K\}} \hat{f}(\boldsymbol{w}_1, \cdots, \boldsymbol{w}_K) = \frac{1}{K} \sum_{k=1}^{K} \left\{ \hat{F}_k(\boldsymbol{w}_k) := \mathbb{E}[\mathcal{L}_{(\boldsymbol{x},y) \sim \mathcal{D}_k}(\boldsymbol{w}_k; (\boldsymbol{x}, y))] \right\},$$

according to (Zhang et al., 2020; Hanzely et al., 2021). The problem could be separately solved by individual client with no communication. However, if the local data is insufficient, poor performance could be observed by this direct solution, since the local models cannot be boosted by other clients.

**Regularized PFL problem.** Regularized PFL can ease the heterogeneous challenges encountered by general FL, while escaping the curse of insufficient samples encountered by the ultimate PFL problem. Inspired by (Chen & Chao, 2021), we formulate the Regularized PFL problem as follows:

$$\text{(P3)} \qquad \min_{\{\boldsymbol{w}_1,\cdots,\boldsymbol{w}_K\}} \bar{f}(\boldsymbol{w}_1,\cdots,\boldsymbol{w}_K) = \frac{1}{K}\sum_{k=1}^{K}\left\{\bar{F}_k\left(\boldsymbol{w}_k\right):=\mathbb{E}[\mathcal{L}_{(\boldsymbol{x},y)\sim\mathcal{D}_k}(\boldsymbol{w}_k;(\boldsymbol{x},y))]\right\}+\mathcal{R}\left(\cdot\right),$$

where $\boldsymbol{w}_k$ denote the personalized models and $\mathcal{R}(\cdot)$ is the regularizer that enables information exchange between clients, making the problem tractable even the local samples are insufficient. However, it remains controversial about how to define the regularizer. Also, the gap between regularized PFL problem (P3) and the ultimate PFL problem (P2) still remains unspecified in existing PFL study.

In this work, our ultimate goal is to solve problem (P2), which requires information exchange between clients to ensure effective solution. Below, instead of utilizing regularizer as in (P3), we alternatively propose a novel Sparse PFL (SPFL) problem to reach the same goal.

**Sparse PFL problem.** By introducing personalized masks into FL, we derive the SPFL problem as

$$\text{(P4)} \qquad \min_{\boldsymbol{w}} f(\boldsymbol{w}) = \frac{1}{K}\sum_{k=1}^{K}\left\{F_k(\boldsymbol{m}_k^*\odot\boldsymbol{w}):=\mathbb{E}[\mathcal{L}_{(\boldsymbol{x},y)\sim\mathcal{D}_k}(\boldsymbol{m}_k^*\odot\boldsymbol{w};(\boldsymbol{x},y))]\right\},$$

where $\boldsymbol{m}_k^*\in\{0,1\}^d$ is a personalized sparse binary mask for $k$-th client. $\odot$ denotes the Hadamard product for two given vectors. Our ultimate goal is to find a global model $\boldsymbol{w}$, such that the personalized model for $k$-th client can be extracted from the global model by personalized mask $\boldsymbol{m}_k^*$, i.e., $\boldsymbol{m}_k^*\odot\boldsymbol{w}$. The element of $\boldsymbol{m}_k^*$ being 1 means that the weight in the global model is active for $k$-th personalized model, otherwise, remains dormant. Thus, the information exchange between all personalized models is enforced by a shared global model $\boldsymbol{w}$.

Compared with existing PFL algorithms, solving our SPFL problem (P4) does not sacrifice additional computation and storage overhead of clients, since we do not maintain both personalized local models and global model in clients as in (Li et al., 2021; Mansour et al., 2020). On contrary, the solution to our problem could potentially lower the communication and computation overhead. Moreover, SPFL problem (P4) can be applied to most of the model architectures without model-specific hyper-parameter tuning, since we do not make model-specific separation of the public and personalized layer in (Arivazhagan et al., 2019; Liang et al., 2020; Collins et al., 2021), or domain-specific fine-tuning in (Chen et al., 2020; Yang et al., 2020).

## 4 FEDSPA: SOLUTION FOR SPFL

In this section, we first introduce our proposed FedSpa in Algorithm 1. Then, we specify the update rule of global model, and two sparse-to-sparse mask searching methods that can be plug in the update process. At last, we give a theoretical analysis on evaluating the quality of the iterates of FedSpa with respect to the ultimate PFL problem (P2).

### 4.1 GLOBAL MODEL UPDATE RULE FOR FEDSPA

**Data Parallel-based Update.** We first propose the following iterative update to solve problem (P4)

$$\boldsymbol{w}_{t+1} = \boldsymbol{w}_t - \frac{\eta}{K}\sum_{k=1}^{K}\boldsymbol{m}_k^*\odot\nabla_{\tilde{\boldsymbol{w}}_{k,t}}\mathcal{L}(\tilde{\boldsymbol{w}}_{k,t};\xi_{k,t}), \qquad (1)$$

where $\xi_{k,t}$ is a batch of data that is uniformly sampled from the $k$-th client's local distribution $\mathcal{D}_k$, $\eta$ is the learning rate, and $\tilde{\boldsymbol{w}}_{k,t} = \boldsymbol{m}_k^*\odot\boldsymbol{w}_t$ is the sparse weights sparsified by mask $\boldsymbol{m}_k^*$. However,

the optimal personalized masks $\{\boldsymbol{m}_k^*\}$ are generally not accessible to us in the solution process. Let $\boldsymbol{m}_{k,t}$ be an intermediate surrogate personalized mask of $\boldsymbol{m}_k^*$, we rewrite Eq. (1) as:

$$\boldsymbol{w}_{t+1} = \boldsymbol{w}_t - \frac{\eta}{K} \sum_{k=1}^{K} \boldsymbol{m}_{k,t} \odot \nabla_{\tilde{\boldsymbol{w}}_{k,t}} \mathcal{L}(\tilde{\boldsymbol{w}}_{k,t}; \xi_{k,t}). \tag{2}$$

For our proposed update rule, it is worth mentioned that: (**i**) Some coordinates of the model weights have been made zero before doing the forward process, i.e., not all the parameters have to be involved when calculating $\mathcal{L}(\tilde{\boldsymbol{w}}_{k,t}; \xi_{k,t})$. This means that the computation overhead in the forward process could be potentially saved. (**ii**) In the backward process, the stochastic gradient $\nabla_{\tilde{\boldsymbol{w}}_{k,t}} \mathcal{L}(\tilde{\boldsymbol{w}}_{k,t}; \xi_{k,t})$ is masked again by $\boldsymbol{m}_{k,t}$, which means that we do not need to backward the gradient for those sparse coordinates. Thus, the computation cost can be largely saved.

**FL-adapted Update.** To save the communication overhead, we integrate the idea from *local SGD* (Stich, 2018) and *partial participation* to our solution. Let $\tilde{\boldsymbol{w}}_{k,t,\tau}$ denote the weights before doing $(\tau+1)$-th step of local SGD and set $\tilde{\boldsymbol{w}}_{k,t,0} = \boldsymbol{m}_{k,t} \odot \boldsymbol{w}_t$ (i.e., the local weights will be synchronized every $N$ steps with the global weights). Then, each client $k \in S_t$ updates its model as below:

$$\tilde{\boldsymbol{w}}_{k,t,\tau+1} = \tilde{\boldsymbol{w}}_{k,t,\tau} - \eta \boldsymbol{m}_{k,t} \odot \nabla_{\tilde{\boldsymbol{w}}_{k,t,\tau}} \mathcal{L}(\tilde{\boldsymbol{w}}_{k,t,\tau}; \xi_{k,t,\tau}), \tau = 0, 1, \ldots, N-1, \tag{3}$$

where $\xi_{k,t,\tau}$ is a batch of sampled data in $\tau$-th step at round $t$. After the local training is finished, the models of participated clients are updated and aggregated to the global model in server as follows:

$$\boldsymbol{w}_{t+1} = \boldsymbol{w}_t - \frac{1}{|S_t|} \sum_{k \in S_t} (\tilde{\boldsymbol{w}}_{k,t,0} - \tilde{\boldsymbol{w}}_{k,t,N}), \tag{4}$$

where $S_t$ is the set of clients selected to be participant in round $t$. According to Eq. (3), the update synchronized to the server (i.e., $\tilde{\boldsymbol{w}}_{k,t,0} - \tilde{\boldsymbol{w}}_{k,t,N}$), and the model distributed to clients (i.e., $\tilde{\boldsymbol{w}}_{k,t,0}$) are all sparse with a constant sparsity. Therefore, the communication overhead over synchronization could be largely saved. At last, we summarize our proposed FedSpa in Algorithm 1.

---

**Algorithm 1** FedSpa

**Input** Training iteration $T$; Learning rate $\eta$; Local Steps $N$; Random seed *seed*;

1: **procedure** SERVER'S MAIN LOOP
2:     Randomly initialize global model $\boldsymbol{w}_0$
3:     $\boldsymbol{m}_{k,0} = $ Initialization(*seed*) for $k = 0, 1, \ldots, K$     ▷ Initialize masks via a mask-searching solution
4:     **for** $t = 0, 1, \ldots, T-1$ **do**
5:         Uniformly sample a fraction of client into $S_t$
6:         **for** each client $k \notin S_t$ **do**
7:             $\boldsymbol{m}_{k,t+1} = \boldsymbol{m}_{k,t}$     ▷ Inherit masks for round $t+1$ if not chosen
8:         **for** $k \in S_t$ **do**
9:             Send $\tilde{\boldsymbol{w}}_{k,t,0} = \boldsymbol{m}_{k,t} \odot \boldsymbol{w}_t$ to client $k$
10:           Call Client $k$'s main loop and receive $\boldsymbol{U}_{k,t}$ and $\boldsymbol{m}_{k,t+1}$
11:         $\boldsymbol{w}_{t+1} = \boldsymbol{w}_t - \frac{1}{S} \sum_{k \in S_t} \boldsymbol{U}_{k,t}$     ▷ Average and apply the update
12: **procedure** CLIENT'S MAIN LOOP
13:     **for** $\tau = 0, 1, \ldots, N-1$ **do**
14:         Sample a batch of data $\xi_{k,t,\tau}$ from local dataset
15:         $\boldsymbol{g}_{k,t,\tau}(\tilde{\boldsymbol{w}}_{k,t,\tau}) = \nabla_{\tilde{\boldsymbol{w}}_{k,t,\tau}} \mathcal{L}(\tilde{\boldsymbol{w}}_{k,t,\tau}; \xi_{k,t,\tau})$
16:         $\tilde{\boldsymbol{w}}_{k,t,\tau+1} = \tilde{\boldsymbol{w}}_{k,t,\tau} - \eta \boldsymbol{m}_{k,t} \odot \boldsymbol{g}_{k,t,\tau}(\tilde{\boldsymbol{w}}_{k,t,\tau})$     ▷ Local update with fixed mask $\boldsymbol{m}_{k,t}$
17:     $\boldsymbol{U}_{k,t} = \tilde{\boldsymbol{w}}_{k,t,0} - \tilde{\boldsymbol{w}}_{k,t,N}$
18:     $\boldsymbol{m}_{k,t+1} = $ Next_Masks($\cdot$)     ▷ Plug in a mask-searching solution to produce next masks
19:     Send back $\boldsymbol{U}_{k,t}$ and $\boldsymbol{m}_{k,t+1}$

---

## 4.2 SPARSE-TO-SPARSE MASK SEARCHING TECHNIQUE

The framework of FedSpa is extensible. Specifically, we use the mask surrogate $m_{k,t}$ in Eq. (3) to perform the update, which allows us to plug in arbitrary mask searching techniques to determine

the iterating process of $m_{k,t}$. In this work, we nominate two kinds of sparse-to-sparse training techniques: modified dynamic sparse training (DST) (Liu et al., 2021) and Random Static Masks (RSM), into FedSpa to reduce the computation and communication cost.

**Modified DST for FedSpa.** Our modified DST solution (see Algorithm 2) for FL follows these procedures. Firstly, randomly initialize the same mask for each client based on Erdós-Rényi Kernel (ERK) (Evci et al., 2020). Secondly, after local training, each client prunes out a number of unpruned weights with the smallest magnitude, and the number of weights being pruned is determined by a decayed pruned rate. Thirdly, recover the same amount of weights pruned in the last step. We follow the recovery process as in (Evci et al., 2020) by utilizing the gradient information to do the recovery. By our DST method, the number of sparse weights (aka. *sparse volume*) remains a constant (i.e., $\beta$) throughout the whole training process.

---

**Algorithm 2** Modified DST for FedSpa

**Input** Initial pruning rate $\alpha_0$; Set of Model layers $\mathcal{J}$;
1: **procedure** INITIALIZATION(*seed*)
2:    Randomly initialize $m_{k,0}$ using the same random seed *seed*.
3: **procedure** NEXT_MASKS($\tilde{w}_{k,t,N}$)
4:    Decay $\alpha_t$ using cosine annealing with initial pruning rate $\alpha_0$
5:    Sample a batch of data and backward the dense gradient $g(\tilde{w}_{k,t,N})$
6:    **for** layer $j \in \mathcal{J}$ **do**
7:       Update mask $m_{k,t+\frac{1}{2}}^{(j)}$ by zeroing out $\alpha_t$-proportion of weights with magnitude pruning
8:       Update mask $m_{k,t+1}^{(j)}$ via recovering weights with gradient information $g(\tilde{w}_{k,t,N})$
9:    Return $m_{k,t+1}$

---

*Remark.* We highlight our main modification over traditional DST techniques like Rigl (Evci et al., 2020) and Set (Mocanu et al. (2018)) to an FL context exist in two main aspects: (**i**) The pruning is performed individually by each client based on their local models, and the gradient used for weights recovery is derived using the client's local training data. (**ii**) Once the next masks are generated, existing DST solutions immediately apply them to the local model weight. Indicated by Liu et al. (2021), by doing so, the recovered coordinate may need extra training steps to grow from 0 to a dense value. Our solution relieves this problem by applying the new mask on the global weights (which are dense), such that the recovered coordinates could have a dense initial value to warm-start.

**RSM for FedSpa.** RSM (shown in Algorithm 3) is basically fixing $m_{k,t}$ for all $k \in [K]$ to the same randomly initialized mask, which remains unchanged during the whole training session. This solution also ensures the same sparse volume for all the clients throughout the training process, and could also reduce the computation and communication overhead as DST. Interestingly, within the setting of the homogeneous data distribution, we empirically show that RSM is more effective than DST in FedSpa.

---

**Algorithm 3** RSM for FedSpa

1: **procedure** INITIALIZATION(*seed*)
2:    Randomly initialize $m_{k,0}$ using the same random seed *seed*
3: **procedure** NEXT_MASKS
4:    $m_{k,t+1} = m_{k,t}$
5:    Return $m_{k,t+1}$

---

### 4.3 THEORETICAL ANALYSIS

In this section, we shall introduce the convergence property of personalized models produced by FedSpa. We first give the following assumptions.

**Assumption 1** (Coordinate-wise bounded gradient dissimilarity)**.** *For any $\tilde{w} \in \mathbb{R}^d$, there exists a constant $G \geq 0$ bounding the coordinate-wise gradient dissimilarity over all clients, i.e.,* $\left\| \nabla F_k(\tilde{w}) - \frac{1}{K} \sum_{k'=1}^{K} \nabla F_{k'}(\tilde{w}) \right\|_\infty \leq G.$

**Assumption 2** (Bounded variance)**.** *Assume that $g_{k,t,\tau}(\tilde{w}) := \nabla \mathcal{L}(\tilde{w}; \xi_{k,t,\tau})$ is an unbiased estimator of $\nabla F_k(\tilde{w})$ with bounded variance, i.e., $\mathbb{E}\left[ \|g_{k,t,\tau}(\tilde{w}) - \nabla F_k(\tilde{w})\|^2 \right] \leq \sigma^2, \forall k, t, \tau, \tilde{w} \in \mathbb{R}^d.$*

**Assumption 3** (L-smoothness)**.** *We assume L-smoothness over the client's loss function, i.e., $\|\nabla F_k(\tilde{w}_1) - \nabla F_k(\tilde{w}_2)\| \leq L\|\tilde{w}_1 - \tilde{w}_2\|$ holds for arbitrary $\tilde{w}_1, \tilde{w}_2 \in \mathbb{R}^d.$*

**Assumption 4** (Coordinate-wise bounded gradient). *Suppose the coordinate-wise gradient of each client is upper-bounded, i.e., $\|\nabla_{\tilde{\boldsymbol{w}}} F_k(\tilde{\boldsymbol{w}})\|_\infty \leq B$.*

Assumptions 2 and 3 are commonly used for characterizing the convergence of FL algorithms. We modify the other two assumptions slightly from their counterparts in existing FL literature (Xu et al., 2021, see Assumptions 3 and 5) by replacing the $L_2$ norm with $L_\infty$ norm. These modifications are made to reveal the coordinate-wise gradient information in our analysis.

**Theorem 1** (Convergence of personalized models). *Given the above assumptions, suppose the learning rate satisfies $\eta \leq \frac{1}{16LN}$, the optimal personalized masks $\boldsymbol{m}_k^* \in \{0,1\}^d$ and the evolutionary masks $\boldsymbol{m}_{k,t} \in \{0,1\}^d$ maintain the same sparse volume constraint, i.e., $\|\mathbf{1} - \boldsymbol{m}_k^*\|_0 = \|\mathbf{1} - \boldsymbol{m}_{k,t}\|_0 = \beta$, the personalized sparse models $\{\tilde{\boldsymbol{w}}_{k,t}\}$ generated via FedSpa exhibits the following convergence property towards the optimal solution of ultimate PFL problem (P2):*

$$\frac{1}{TK} \sum_{t=0}^{T-1} \sum_{k=1}^{K} \mathbb{E}\left[\|\nabla F_k(\tilde{\boldsymbol{w}}_{k,t})\|^2\right] \leq \frac{3(f(\boldsymbol{w}_0) - f(\boldsymbol{w}^*))}{T\eta N\kappa} + \frac{3}{T} \sum_{t=0}^{T-1} \rho_t + \Upsilon, \qquad (5)$$

*where $\kappa = \frac{1}{2} - 150N^3\eta^3 L^3 - 15N^2\eta^2 L^2 - 5N\eta L$ and $\rho_t = (25N^3\eta^4 L^3 + \frac{5N^2\eta^3 L^2}{2})(\sigma^2 + 18N\Phi_t) + \frac{4N^2\eta^2 L + N\eta}{2K} \sum_k dist(\boldsymbol{m}_{k,t}, \boldsymbol{m}_k^*)B^2 + 9N^2\eta^2 L\Phi_t + \frac{N\eta^2 L\sigma^2}{S}$, $\Phi_t = \frac{1}{K} \sum_k (\underbrace{(d-\beta)G^2}_{\approx 0 \; if \; \beta \to d} + \frac{1}{K} \sum_{k'} B^2(\underbrace{dist(\boldsymbol{m}_{k,t}, \boldsymbol{m}_{k',t})}_{\approx 0 \; if \; \boldsymbol{m}_{k,t} \to \boldsymbol{m}_{k',t}} + \underbrace{dist(\boldsymbol{m}_{k',t}, \boldsymbol{m}_{k'}^*)}_{\approx 0 \; if \; \boldsymbol{m}_{k',t} \to \boldsymbol{m}_{k'}^*}))$, $\Upsilon = \underbrace{3(d-\beta)G^2}_{\approx 0 \; if \; \beta \to d} + \underbrace{3\beta B^2}_{\approx 0 \; if \; \beta \to 0} + \frac{3}{K^2} \sum_k \sum_{k'} \underbrace{dist(\boldsymbol{m}_k^*, \boldsymbol{m}_{k'}^*)B^2}_{\approx 0 \; if \; \boldsymbol{m}_k^* \to \boldsymbol{m}_{k'}^*}$, $dist(\boldsymbol{m}_1, \boldsymbol{m}_2)$ is the hamming distance[2] between masks $\boldsymbol{m}_1$ and $\boldsymbol{m}_2$.*

Below, we give several comments on the above theorem. We focus our analysis on the on the second and third residuals, which do not vanish as communication round $T \to \infty$

- **Impact of sparse volume.** The result in Eq. (5) is highly related to setting of sparse volume, i.e., $\beta$. The term $\beta B^2$ would vanish as $\beta \to 0$, while the term $(d-\beta)G^2$ would disappear as $\beta \to d$. When the data is highly heterogeneous, i.e., gradient dissimilarity is drastic, $G$ could be prohibitively large to dominate the error. Then setting the sparsity level to a relatively large value, which results in a lower $(d-\beta)G^2$, would potentially give a smaller non-vanished error. On contrary, when the data heterogeneity is mild, the error could be dominated by $\beta B^2$. Then setting an excessively high sparsity might even hurt the performance. This conclusion is also evidenced by our experimental results (see Figure 1 in Section 1, where the optimal sparsity setting of FedSpa that achieves the highest accuracy is 0.5).

- **Error by evolutionary mask searching.** Recall that FedSpa uses evolutionary masks as a surrogate of optimal masks to perform update on the global model $w_t$ (see Eq. (3)). This replacement may bring additional errors to the convergence bound, as the hamming distance between the evolutionary masks and optimal masks (i.e., the term $dist(\boldsymbol{m}_{k,t}, \boldsymbol{m}_k^*)$) exists in our bound.

- **Error by dissimilar evolutionary masks.** The theoretical result on the term $dist(\boldsymbol{m}_{k,t}, \boldsymbol{m}_{k',t})$ in $\Phi_t$ indicates that the dissimilarity among the evolutionary masks may also play a role in optimizing the ultimate PFL problem (P2). This term could be minimized if the evolutionary masks remain the same throughout the training process, and would dominate the error if the masks are too distinct. This observation exactly motivates us to present the naive mask searching technique RSM in Algorithm 3, which forces all the clients to share the same random masks.

## 5 EXPERIMENTS

In this section, we conduct extensive experiments on verifying the efficacy of the proposed FedSpa. Our implementation of FedSpa is based on an open-source FL simulator FedML (He et al., 2020).

### 5.1 EXPERIMENTAL SETUP

**Dataset.** We evaluate the efficacy of FedSpa on EMNIST-Letter (EMNIST-L henceforth), CIFAR10, and CIFAR100 datasets. We simulate the client's data distribution on Non-IID and IID setting. We

---

[2]Hamming distance measures the number of positions at which the two masks are different.

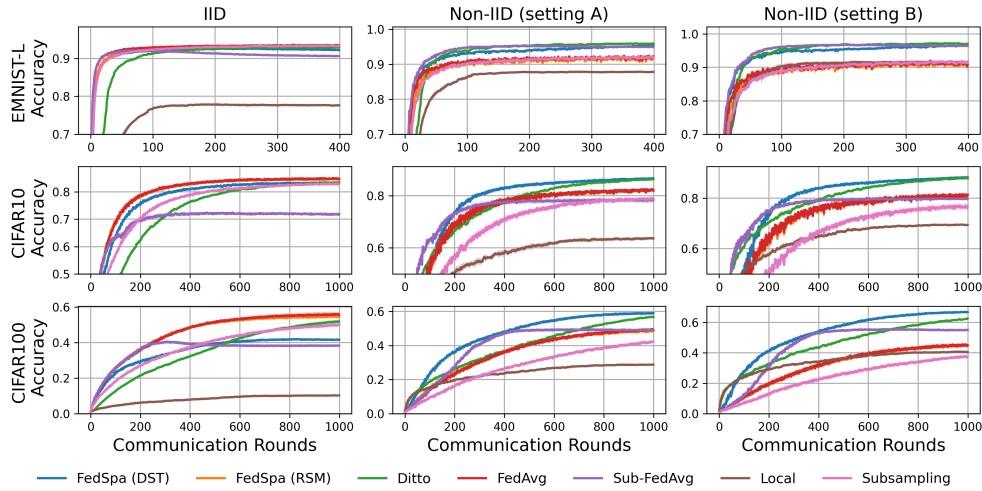

Figure 3: Test Accuracy vs. Communication Rounds

simulate two groups of Non-IID settings via $\gamma$-Dirichlet distribution, named setting A and setting B. Setting A and setting B respectively specify $\gamma = 0.2, 0.1$ for both EMNIST-L and CIFAR100, while specify $\gamma = 0.5, 0.3$ for CIFAR10. Details of our simulation setting are available in Appendix B.1.

**Baselines.** We compare our proposed FedSpa with four baselines, including FedAvg (McMahan et al. (2016)), Sub-FedAvg (Vahidian et al. (2021)), Ditto (Li et al. (2021)) and Local. We tune the hyper-parameters of the baselines to their best states. Specifically, the regularization factor of Ditto is set to 0.5. The prune rate each round, distance threshold, and accuracy threshold of Fed-Subavg are fixed to 0.05, 0.0001, 0.5, respectively. We ran 3 random seeds in our comparison.

**Models and hyper-parameters.** We use LeNet5 for EMNIST-L, VGG11 for CIFAR10, and ResNet18 for CIFAR100 in our experiment. We use a SGD optimizer with weight decayed parameter 0.0005. The learning rate is initialized with 0.1 and decayed with 0.998 after each communication round. We simulate 100 clients in total, and in each round 10 of them are picked to perform local training (the setting follows McMahan et al. (2016)). For all the methods except Ditto, local epochs are fixed to 5. For Ditto, in order to ensure a fair comparison, each client uses 3 epochs for training of the local model, and 2 epochs for global model training. The batch size of all the experiments is fixed to 128. For FedSpa, the pruning rate (i.e., $\alpha_t$) is decayed using cosine annealing with an initial pruned rate 0.5. The initial sparsity of layers is initialized by ERK with scale parameter 1.

## 5.2 MAIN PERFORMANCE EVALUATION

We fix the dense ratio of FedSpa (DST), FedSpa (RSM), and the final dense ratio of Fed-SubAvg both to 0.5 (i.e., 50% of parameters are pruned) in our main evaluation. Other hyper-parameters are fixed as default. Figure 3 and Table 1 illustrate the training performance of different algorithms on three datasets. We evaluate the performance based on the following metrics:

**Final Accuracy.** In the Non-IID setting, we show that FedSpa (DST) achieves remarkable performance. Specifically, in Non-IID setting B of CIFAR100, FedSpa (DST) achieves respectively 4.4%, 11.9% and 21.9% higher final model accuracy, compared with Ditto, Sub-FedAvg and FedAvg. FedSpa (DST) seems to achieve better performance as the FL tasks becoming difficult (since better performance is observed in a higher Non-IID extent, and in datasets that are intrinsically more difficult). Interestingly, in the IID setting, we show that all the personalized solutions exhibit some extents of performance degradation, which become more significant as the dataset becomes challenging. The compression-based methods seem to be especially vulnerable in this setting. Our interpretation for this phenomenon is that: since the information exchange between clients would be limited by employing different sub-networks for training, the clients could not efficiently make an effective fusion on their models through parameter averaging. This hypothesis is substantiated by our experiment on FedSpa (RSM), an alternative implementation of FedSpa, which forces all the masks to maintain the same sub-network. FedSpa (RSM) achieves commensurate performance with FedAvg in the IID setting, outperforming the personalized solutions. The reason FedSpa (DST)

Table 1: Table illustrating performance of different methods.

| Task | Method | IID | | | Non-IID | | | | | |
| | | | | | Setting A | | | Setting B | | |
| | | Acc | Comm Cost (GB) | FLOPs (1e14) | Acc | Comm Cost (GB) | FLOPs (1e16) | Acc | Comm Cost (GB) | FLOPs (1e14) |
|---|---|---|---|---|---|---|---|---|---|---|
| EMNIST-L (LeNet) | FedSpa (DST) | 92.2±0.1 | **7.0** | 2.0 | 95.3±0.1 | **7.0** | 2.0 | 96.5±0.2 | **7.0** | 2.0 |
| | FedSpa (RSM) | 92.9±0.1 | **7.0** | 2.0 | 91.9±0.2 | **7.0** | 2.0 | 90.6±0.9 | **7.0** | 2.0 |
| | Ditto | 92.9±0.1 | 14.1 | 3.5 | **95.9**±0.1 | 14.1 | 3.5 | **97.0**±0.2 | 14.1 | 3.5 |
| | FedAvg | **93.5**±0.2 | 14.1 | 3.5 | 92.3±0.3 | 14.1 | 3.5 | 90.9±0.8 | 14.1 | 3.5 |
| | Sub-FedAvg | 90.7±0.2 | 9.5 | **1.9** | 94.9±0.2 | 9.4 | **1.9** | 96.4±0.2 | 9.4 | **1.9** |
| | Local | 77.6±0.3 | - | 3.5 | 87.8±0.1 | - | 3.5 | 91.6±0.5 | - | 3.5 |
| | Subsampling | 93.3±0.2 | 10.5 | 3.5 | 92.0±0.4 | 10.5 | 3.5 | 91.3±0.6 | 10.5 | 3.5 |
| CIFAR-10 (VGG11) | FedSpa (DST) | 83.4±0.1 | **369.2** | 172.9 | **86.6**±0.5 | **369.2** | 173.3 | **88.2**±0.4 | **369.2** | 173.5 |
| | FedSpa (RSM) | 84.5±0.1 | **369.2** | 172.9 | 82.1±0.2 | **369.2** | 173.3 | 80.9±0.2 | **369.2** | 173.5 |
| | Ditto | 83.5±0.2 | 738.5 | 229.3 | 86.4±0.6 | 738.5 | 229.8 | 87.8±0.3 | 738.5 | 230.0 |
| | FedAvg | **84.8**±0.3 | 738.5 | 229.3 | 82.0±0.4 | 738.5 | 229.8 | 81.4±0.4 | 738.5 | 230.0 |
| | Sub-FedAvg | 71.8±0.3 | 410.2 | **121.4** | 78.3±1.0 | 424.7 | **120.6** | 79.6±0.6 | 416.9 | **119.8** |
| | Local | 42.5±0.2 | - | 229.3 | 63.6±0.6 | - | 229.8 | 69.4±0.2 | - | 230.0 |
| | Subsampling | 83.0±0.4 | 553.9 | 229.3 | 78.9±0.5 | 553.9 | 229.8 | 76.7±1.0 | 553.9 | 230.0 |
| CIFAR-100 (ResNet18) | FedSpa (DST) | 41.5±0.5 | **448.8** | 705.1 | **59.0**±1.0 | **448.8** | 704.9 | **66.9**±0.2 | **448.8** | 704.8 |
| | FedSpa (RSM) | 54.6±1.1 | **448.8** | 705.1 | 48.7±0.5 | **448.8** | 704.9 | 64.6±0.5 | **448.8** | 704.8 |
| | Ditto | 51.9±1.1 | 897.6 | 833.2 | 56.8±0.6 | 897.6 | 833.0 | 62.5±0.2 | 897.6 | 832.9 |
| | FedAvg | **55.7**±1.3 | 897.6 | 833.2 | 49.3±0.4 | 897.6 | 833.0 | 45.0±0.9 | 897.6 | 832.9 |
| | Sub-FedAvg | 38.3±0.8 | 616.5 | **494.1** | 49.2±0.7 | 624.4 | 508.4 | 55.0±0.7 | 612.8 | 496.1 |
| | Local | 10.3±0.3 | - | 833.2 | 28.8±0.1 | - | 833.0 | 40.5±0.4 | - | 832.9 |
| | Subsampling | 49.8±1.3 | 673.2 | 833.2 | 42.3±0.8 | 673.2 | 833.0 | 37.6±1.1 | 673.2 | 832.9 |

performing better in Non-IID setting can also be explained by Theorem 1, which concludes that in a Non-IID setting, setting proper sparsity for personalized models may promise a better convergence.

**Convergence.** As shown in Table 2, FedSpa achieves significantly faster convergence, which potentially saves the communication rounds to train a model from scratch to a specific accuracy.

Table 2: Communication rounds to a fixed accuracy.

| CIFAR10 | IID | | | Non-IID | | | | | |
| | | | | Setting A | | | Setting B | | |
| | Acc@70 | Acc@75 | Acc@80 | Acc@70 | Acc@75 | Acc@80 | Acc@70 | Acc@75 | Acc@80 |
|---|---|---|---|---|---|---|---|---|---|
| FedSpa (DST) | 134.0±2.9 | 183.3±6.8 | 312.3±16.2 | 167.3±4.0 | 210.3±4.2 | 281.3±19.1 | 164.3±5.0 | 206.3±4.1 | **270.0**±5.1 |
| FedSpa (RSM) | **101.3**±1.7 | 141.3±6.2 | 237.0±6.4 | 195.3±10.7 | 271.3±16.2 | 471.7±19.8 | 252.0±12.7 | 339.0±20.6 | 614.0±72.8 |
| Ditto | 284.7±8.1 | 370.3±9.3 | 549.3±22.6 | 242.3±12.7 | 334.0±16.5 | 466.3±30.3 | 190.3±6.1 | 278.0±22.0 | 417.7±10.2 |
| FedAvg | 105.0±2.2 | **140.3**±4.7 | **228.7**±23.5 | 198.3±14.8 | 256.7±10.8 | 474.7±31.4 | 241.0±3.7 | 327.3±8.5 | 583.7±65.5 |
| Sub-FedAvg | 197.7±22.9 | > 1000 | > 1000 | **151.7**±10.6 | 235.0±17.1 | > 1000 | **137.3**±1.7 | **191.7**±6.6 | > 1000 |
| Subsampling | 198.0±4.5 | 268.3±4.5 | 457.0±13.5 | 365.3±23.8 | 523.0±43.4 | > 1000 | 466.3±15.0 | 722.7±105.1 | > 1000 |
| CIFAR100 | Acc@40 | Acc@50 | Acc@55 | Acc@40 | Acc@50 | Acc@55 | Acc@40 | Acc@50 | Acc@55 |
| FedSpa (DST) | 536.3±35.9 | > 1000 | > 1000 | **236.3**±12.3 | 442.0±16.9 | 595.0±41.3 | 181.3±7.8 | 314.7±17.4 | 407.7±16.7 |
| FedSpa (RSM) | **239.3**±4.1 | 435.7±25.8 | > 1000 | 460.7±12.5 | > 1000 | > 1000 | 594.0±10.7 | > 1000 | > 1000 |
| Ditto | 545.7±19.4 | 868.7±56.1 | > 1000 | 455.3±6.9 | 724.0±20.2 | 894.0±25.0 | 301.3±10.8 | 534.0±11.3 | 678.7±7.9 |
| FedAvg | 245.0±5.1 | 436.3±25.3 | > 1000 | 470.7±25.4 | > 1000 | > 1000 | 589.7±46.7 | > 1000 | > 1000 |
| Sub-FedAVG | > 1000 | > 1000 | > 1000 | 280.7±2.5 | > 1000 | > 1000 | 246.3±10.1 | 335.3±14.1 | 511.0±85.9 |
| Subsampling | 460.7±16.6 | > 1000 | > 1000 | 845.3±59.2 | > 1000 | > 1000 | > 1000 | > 1000 | > 1000 |

**Training FLOPs and Communication.** From Table 1, FedSpa (DST) achieves 15.4%∼42.9% lower FLOPs than the dense solutions (e.g., Ditto, FedAvg), 13.0%∼28.2% lower communication overhead than another model compression solution Sub-FedAvg, and 50% lower communication than the dense solution. The edge of FedSpa (DST) stems from its training pattern – it is trained from a sparse model, with constant sparsity throughout the training process. However, it is interesting to see that the training FLOPS of Fed-SubAvg is considerably lower than FedSpa, even under the same sparsity setting. This phenomenon stems from our ERK initialization, which is essential for the high performance of our solution, for which we will have a further discussion in Appendix B.4.

## 6 CONCLUSIONS

In this paper, we propose FedSpa, a personalized FL solution that enables sparse-to-sparse training and efficient sub-model aggregation. As demonstrated by our experiments, FedSpa exhibits outstanding performance in the Non-IID setting, outperforming other existing solutions in terms of accuracy, convergence speed as well as communication overhead. Additionally, we present theoretical analysis to evaluate the error bound of FedSpa towards the ultimate PFL problem. Future direction includes designing new model aggregation solutions for the sparse sub-network, and new mask searching techniques specifically targeting on federated learning process.

## 7 REPRODUCIBILITY STATEMENT

For sake of reproducibility of our solution, we make the following efforts: (**i**) In Appendix B.1, we clearly state the data splitting method for IID and Non-IID data distribution. (**ii**) In Appendix B.2, the network models we used in our experiment are clearly described. (**iii**) In Appendix B.4, the proposed FedSpa with various hyper-parameters during the implementation are also systemically tested to demonstrate its stability and superiority. (**iv**) In Appendix C, we give the self-contained proof of Theorem 1. (**v**) At last, the source-code of FedSpa is enclosed in the supplementary material.

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

# A    CRITICAL COMPONENTS OF FEDSPA (DST)

**ERK initialization.** In Algorithm 2, we use Erdós-Rényi Kernel (ERK) originally proposed by Evci et al. (2020) to initialize the sparsity of each layer. Specifically, the active parameters of the convolutional layer initialized by ERK is proportional to $1\frac{n^{l-1}+n^l+w^l+h^l}{n^{l-1}*n^l*w^l*h^l}$, where $n^{l-1}$, $n_l$ $w^l$ and $h^l$ respectively specify number of input channels, output channels and kernel's width and height in the $l$-th layer. For the linear layer, the number of active parameters scale with $1\frac{n^{l-1}+n^l}{n^{l-1}\div n^l}$ where $n^{l-1}$ and $n^l$ are the number of neurons in the $l-1$-th and $l$-th layer. This initialization basically allows the layer with less parameters have more proportion of active parameters.

**Cosine annealing.** Recall that we set the initial pruning rate as $\alpha_0$ and gradually decay it to 0 with cosine annealing Liu et al. (2021). The update of pruning rate with cosine annealing can be formalized as: $\alpha_t = 0.5 \times \alpha_0 \times \left(1 + \cos\left(\frac{t}{T-1}\pi\right)\right)$. We perform this decay in order to ensure that the network (specifically, its active coordinates) would not experience drastic change on the later stage of training while ensuring that the mask searching is effective on the early stage of training.

# B    DETAILED SETTING AND ADDITIONAL EXPERIMENTS

## B.1    DATA SPLITTING SETTING

In our implementation, we first split the training data (60k pieces of data for CIFAR10 and CI-FAR100, and 145.6k for EMNIST-L, respectively) to clients for IID setting and Non-IID setting. For the IID setting, data are uniformly sampled for each client. For the Non-IID setting, we use $\gamma$-Dirichlet distribution on the label ratios to ensure uneven label distributions among devices as (Hsu et al., 2019). The lower the distribution parameter $\gamma$ is, the more uneven the label distribution will be, and would be more challenging for FL. After the initial splitting of training data, we sample 100 pieces of testing data from the testing set to each client. To simulate the personalized setting, each client's testing data has the same proportion of labels as its training data. Testing of the personalized model is performed by each client based on their personalized data, and the overall testing accuracy is calculated as the average of all the client's testing accuracy. In our experiment, we simulate different Non-IID settings. For CIFAR10, Non-IID setting A and B respectively specify $\gamma = 0.5$ and $\gamma = 0.3$. For EMNIST-L and CIFAR100, since the number of the total labels are bigger[3], we use smaller $\gamma$, wherein setting A and B respectively specify $\gamma = 0.2$ and $\gamma = 0.1$.

## B.2    NETWORK ARCHITECTURES

We follow the Caffe's implementation of LeNet5 [4] (LeCun et al., 2015), VGG11 (Simonyan & Zisserman, 2014) and ResNet18 (He et al., 2016) to do the evaluation. Suggested by Hsieh et al. (2020), DNNs with batch normalization layers (Ioffe & Szegedy, 2015) are particularly vulnerable to the Non-IID setting, suffering significant model quality loss in the FL process. Following the recommendation from Hsieh et al. (2020), we use group normalization (Wu & He, 2018) to substitute the original batch normalization layer in both ResNet18 and VGG11.

## B.3    BASELINE DESCRIPTION

Below, we give a brief introduction of the baselines compared in our evaluations:

- **FedAvg** (McMahan et al., 2016) is the vanilla solution of FL. It utilizes weights average to enable all the clients to collaboratively train a global model, which efficiently absorbs knowledge from personal data resided in clients.
- **Ditto** (Li et al., 2021) is a personalized FL solution aiming to smooth the tension brought by the data heterogeneity problem of FL. Ditto achieves personalization via maintaining both the local models and global model. Specifically, within each round of iteration, each client first trains the global model based on its local empirical loss (which shares the same procedure as FedAvg).

---

[3]26 and 100 labels respectively in EMNIST-L and CIFAR100, while only 10 labels in CIFAR10.
[4]Available in https://github.com/mi-lad/snip/blob/master/train.py

After the global model is updated, each client additionally trains its local model based on a loss function involving its local empirical loss and the proximal term towards the global model. This local training phase is used to extract the global knowledge into each client's local model. Since each client has to maintain and train both local model and global model, Ditto might need extra computation and storage overhead to achieve its personalization.

- **Local** is the direct solution to the ultimate PFL problem (P2). Each client performs SGD based on its local data, and there is no communication between clients. To mimic the FL setting, we sample 10 out of 100 clients to do the local update on its local model after every 5 epochs of training (same with the number of local epochs in a communication round that is performed by other solutions). For sake of consistency, we still use 1 communication round to represent 5 local epochs of Local in our evaluation.

- **Sub-FedAvg** (Vahidian et al., 2021) is a prominent model compression-based PFL. Sub-FedAvg maintains personalized sub-networks for each client. Training of Sub-FedAvg starts from a fully dense model, and this solution iteratively prunes out the parameters and channels as the training progresses. Finally, the commonly shared parameters of each layer are removed, and only the personalized parameters that can represent the features of local data are kept.

- **Subsampling** (Konečnỳ et al., 2016) is a gradient-compression solution aiming to reduce the communication overhead of FL. The local training procedure is the same with FedAvg. The difference is that Subsampling does not communicate the intact model for aggregation, but only communicates the sparse gradient update to the server for aggregation. Explicitly, in each round, the sparse gradient update is produced through element-wisely multiplying a random mask. Different from FedSpa, the randomized mask is independently generated in each round, and would only be used to compress the gradient when uploading the gradient update (which means in the model distribution phase, the model distributed would not be sparsified, and therefore would not save the downlink communication cost).

### B.4 ABLATION STUDY

In this sub-section, we focus on the ablation study of FedSpa. Specifically, we study the impacts of dense ratio, different mask initialization methods, and the gradient-involved weight recovery procedure. Additionally, we present an interesting observation on the performance of the global model trained by our personalized solution. Our ablation study is done with ResNet-18 on CIFAR100.

**Impact of sparsity (aka. sparse ratio).** Fixing other components and hyper-parameters to the default value in our setup, we change the sparsity of FedSpa to 0.2, 0.4, 0.5, 0.6 and 0.8, to show its impact on the algorithm performance. Experimental results are available in Figure 4 and Table 3. By our report, we observe that sparsity may impact learning performance under different data distribution settings. For the IID setting, a higher sparsity seems to seriously degrade the training performance, while for the Non-IID setting, properly sparsifying the model may even enhance the final accuracy, and with a higher Non-IID extent, the benefit of sparsification reinforces. This observation surprisingly coincides with our theoretical conclusion given in Theorem 1, that setting the sparse volume to a proper value when gradient dissimilarity is large (i.e., under a high extent of Non-IID) could foster a better performance of personalized models. But too much sparsity, even in the highly non-iid setting (e.g. $\gamma = 0.1$) leads to performance degradation. On the contrary, while it is iid, the convergence could be dominated by the errors brought by sparsification, and setting the mask to a higher sparsity could possibly enlarge these existing errors.

Another more intuitive interpretation for the impact of sparse ratio is from the perspective of information exchange. Too much sparsification may limit the information exchange between the local sparse models. If all the clients maintain an extremely high sparsity, the intersected coordinates between clients' local sparse models (or identically, their masks) would be small. Then the local update averaging process (see Eq. (4)), the only way to extract global knowledge into the local models, would not be effective. On contrary, while the sparsity is set to an extremely low value, the personalized features of local models could be eliminated, since only limited coordinates in their models are different.

**ERK vs. Uniform sparsity initialization.** Recall our mask initialization procedure in Algorithm 3 that the layer-wise sparsity is initialized by ERK. This in essence ensures that the layer-wise sparsity of a model is scaled with the number of parameters in a layer. Liu et al. (2021) confirms the out-

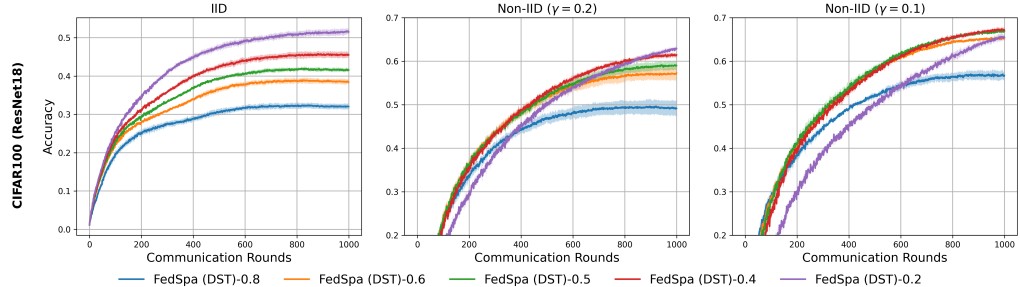

Figure 4: FedSpa (DST) under different sparsity. Numbers in the labels are sparsity.

Table 3: Performance of FedSpa (DST) under different sparsity settings.

| Sparsity | iid | | | Non-iid | | | | | |
|---|---|---|---|---|---|---|---|---|---|
| | | | | $\gamma$=0.2 | | | $\gamma$=0.1 | | |
| | Acc | Comm Cost (GB) | FLOPs (1e16) | Acc | Comm Cost (GB) | FLOPs (1e16) | Acc | Comm Cost (GB) | FLOPs (1e16) |
| 0.2 | **51.5**±0.8 | 718.1 | 8.2 | **62.9**±0.4 | 718.1 | 8.2 | 65.5±0.5 | 718.1 | 8.2 |
| 0.4 | 45.5±0.9 | 538.6 | 7.6 | 61.4±0.6 | 538.6 | 7.5 | **67.2**±0.4 | 538.6 | 7.5 |
| 0.5 | 41.5±0.5 | 448.8 | 7.1 | 59.0±1.0 | 448.8 | 7.0 | 66.9±0.2 | 448.8 | 7.0 |
| 0.6 | 38.4±0.6 | 359.0 | 6.5 | 57.3±1.5 | 359.0 | 6.5 | 65.2±0.2 | 359.0 | 6.5 |
| 0.8 | 32.0±0.7 | **179.5** | **4.6** | 49.2±1.8 | **179.5** | **4.6** | 56.7±0.8 | **179.5** | **4.6** |

standing effect of ERK initialization in improving overall training performance over the centralized training primitive, but it remains unexplored how it performs in our proposed distributed training framework. Below, we show in Figure 5 how accuracy evolves with communication rounds under ERK and Uniform [5] initialization. As shown, a drastic drop of accuracy is observed by replacing ERK with Uniform, by which we conclude that ERK is an essential component for FedSpa (DST).

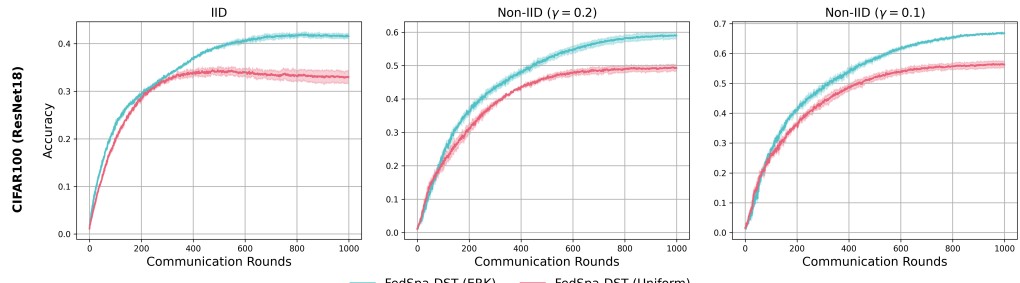

Figure 5: Layer-wise sparsity initialized by ERK or Uniform. Sparsity of FedSpa is fixed to 0.5.

However, though a significant accuracy enhancement is observed, we note that integrating ERK may sacrifice potentially more FLOPS reduction. This observation can be found in Table 6, wherein our results show that initialization with Uniform can save 34.3% FLOPs of that with ERK.

Table 4: Performance of FedSpa under ERK and Uniform initialization.

| Methods | iid | | | Non-iid | | | | | |
|---|---|---|---|---|---|---|---|---|---|
| | | | | $\gamma$=0.2 | | | $\gamma$=0.1 | | |
| | Acc | Comm Cost (GB) | FLOPs (1e16) | Acc | Comm Cost (GB) | FLOPs (1e16) | Acc | Comm Cost (GB) | FLOPs (1e16) |
| ERK | **41.5**±0.5 | 448.8 | 7.1 | **59.0**±1.0 | 448.8 | 7.0 | **66.9**±0.2 | 448.8 | 7.0 |
| Uniform | 33.0±1.4 | 448.8 | **4.6** | 49.3±0.9 | 448.8 | **4.6** | 56.3±1.1 | 448.8 | **4.6** |

**Different or same mask initialization.** Recall that based on the layer-wise sparsity calculated by ERK, FedSpa uses the same random seed to initialize the mask, so as to make the mask exploration

---

[5]Uniform enforces the same sparsity for all the layers in a model.

of all clients started from the same mask. In the following, we give another implementation that allows each client to share different masks in the beginning.

As shown in Figure 6, we surprisingly find that for FedSpa (DST), maintaining different masks in initialization may slightly enhance its training performance in IID and Non-IID ($\gamma = 0.2$) setting. We hypothesize that by different mask initialization, each client could more efficiently search for their optimal masks that better represents the features and labels of the personal data.

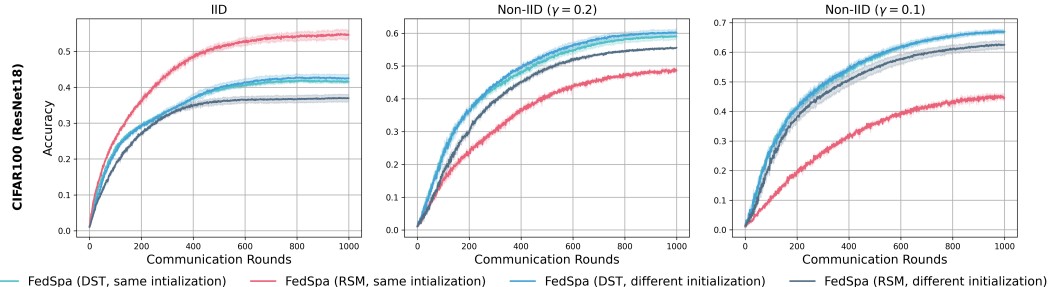

Figure 6: Initialization based on same or different masks. Sparsity of FedSpa is fixed to 0.5.

For FedSpa (RSM), compared with initialization using the same mask, different mask initialization may result in a drastic performance loss in the IID setting, and a significant improvement in the Non-IID setting. With the same mask initialization of RSM, each client consistently trains based on the same sub-network, which completely eliminates personalization. So this setting shares a similar performance with FedAvg – with satisfactory performance in IID setting and rather weak performance in Non-IID setting. On contrary, by initializing different masks in the beginning, FedSpa (RSM) reserves some degrees of personalization, since only the intersected coordinates in their local models are shared and updated by the information exchange (i.e., average) process. Consequently. FedSpa (RSM) with different mask initializations has a similar performance pattern with FedSpa (DST).

Another interesting observation is that FedSpa (RSM) with different mask initialization cannot outperform FedSpa (DST) in both the two groups of Non-IID settings. This indicates that the DST mask searching process is effective to achieve a superior performance of FedSpa in Non-IID setting.

**Weight recovery w/ or w/o gradient information.** Recall that in FedSpa (DST), we proposed to use gradient information to recover the pruned weights, which is empirically proven in (Evci et al., 2020) to outperform its random recovery counterpart in Set (Mocanu et al., 2018). Specifically, for gradient information-based recovery, the weight coordinates with the top-$\alpha_t$ magnitude of the gradient would be recovered, while for random recovery, the coordinates are recovered randomly. To demonstrate the impact of the weight recovery method over FedSpa (DST), in Figure 7, we compare the gradient information-based recovery with random recovery. Our experimental result demonstrates that recovery with gradient information could slightly accelerate the convergence and enhance the final accuracy in our FedSpa framework.

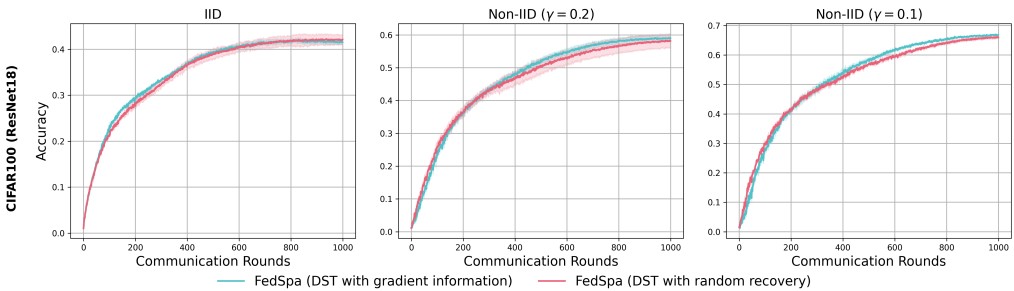

Figure 7: Recovery with gradient information or random recovery. Sparsity is fixed to 0.5.

**Global model vs. Personalized model.** In our main experimental result, all the testings are conducted by clients based on their own personalized models. But it is interesting to evaluate whether

Table 5: Wall time of FedSpa (DST) for local training and mask searching.

| Task | Wall Time (Train) | Wall Time (Mask Search) | Ratio (Mask Search/Train) |
|---|---|---|---|
| EMNIST-LeNet (CPU) | 1.03±0.04s | 0.09±0.0s | 8.92%±0.6 |
| CIFAR10-VGG11 (CPU) | 11.4±0.25s | 2.19±0.11s | 19.19%±1.09 |
| CIFAR100-Resnet18 (CPU) | 28.61±0.38s | 3.7±0.28s | 12.93%±1.08 |
| EMNIST-LeNet5 (GPU) | 0.39±0.01s | 0.02±0.0s | 5.66%±0.25 |
| CIFAR10-VGG11 (GPU) | 1.56±0.03s | 0.22±0.01s | 14.3%±0.48 |
| CIFAR100-Resnet18 (GPU) | 2.71±0.01s | 0.33±0.01s | 12.06%±0.2 |

the global model trained by FedSpa itself could converge, or even could achieve commensurate performance with the global model trained by general FL solution (e.g. FedAvg). As demonstrated by Figure 8, we empirically find that in the IID setting, the global model trained by FedSpa cannot recover the performance of that trained by FedAvg, and a considerable performance drop is also observed in the Non-IID setting. Another observation is that the global model of FedSpa surprisingly maintains roughly the same performance as its personalized models in the IID setting, but conceivably suffers significant performance loss in the Non-IID setting. This corroborates our conclusion that sub-networks extracted from a global model may potentially outperform the full model, under the condition that the data distributions of clients are skewed (or heterogeneous).

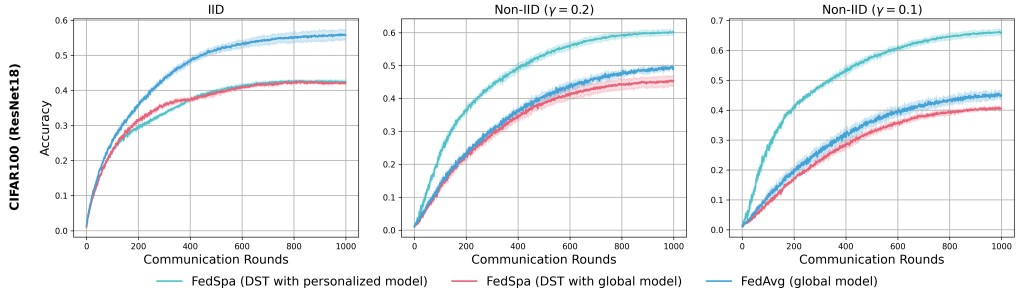

Figure 8: Global model vs. Personalized models. Sparsity of FedSpa is fixed to 0.5.

**Wall time.** Recall that we do an additional mask searching procedure in FedSpa (DST), which might possibly induce extra wall time on the local devices. We show in Table 5 the wall time used for training and mask searching in one single local round. The sparsity used for this experiment is fixed to 0.5, while other parameters remain the default setting (see Section 5.1). We use one single 1080Ti to perform training for the GPU-based experiment, while the CPU-based experiment is conducted on an Intel(R) Xeon(R) CPU E5-2620 v4 @ 2.10GHz with 8 cores. Our experimental results confirm that the mask searching process only accounts for a small portion of wall time (approximately $5\% - 20\%$) for the entire computation time on the local devices.

## C PROOF OF THEOREM 1

In this section, for sake of readability, we first clarify the notations and reiterate several facts that we use in our proof. Then we present several lemmas that are commonly used in the FL literature (see (Karimireddy et al., 2020; Xu et al., 2021)). Later, several key lemmas are listed with exhaustive proof, and finally, the proof of our main theorem is given by exploiting the listed lemmas and facts.

### C.1 NOTATIONS AND FACTS

Throughout the proof, we assume $\sum_{\tau}$ equivalent to $\sum_{\tau=0}^{N-1}$, $\sum_{k}$ equivalent to $\sum_{k=1}^{K}$, and $\sum_{k,\tau}$ equivalent to $\sum_{k=1}^{K}\sum_{\tau=0}^{N-1}$ unless otherwise specified. In our proof, we reuse most of the notations

from our problem formulation part. We use $\boldsymbol{g}_{k,t,\tau}(\tilde{\boldsymbol{w}}_{k,t,\tau}) = \nabla_{\tilde{\boldsymbol{w}}_{k,t,\tau}} \mathcal{L}(\tilde{\boldsymbol{w}}_{k,t,\tau}; \xi_{k,t,\tau})$ to denote the stochastic gradient of client $k$ in round $t$ and at step $\tau$.

Then as per our formulation in Section 4.1, we reiterate the following facts, which would be heavily used in our proof.

**Fact 1** (Local step). *As per Eq. (3), one local step of client's update can be formalized as follows:*

$$\tilde{\boldsymbol{w}}_{k,t,\tau+1} = \tilde{\boldsymbol{w}}_{k,t,\tau} - \eta \boldsymbol{m}_{k,t} \odot \boldsymbol{g}_{k,t,\tau}(\tilde{\boldsymbol{w}}_{k,t,\tau}), \tag{6}$$

*where $\boldsymbol{g}_{k,t,\tau}(\tilde{\boldsymbol{w}}_{k,t,\tau})$ is the stochastic gradient over the sparse model weights $\tilde{\boldsymbol{w}}_{k,t,\tau}$, and $\tilde{\boldsymbol{w}}_{k,t,0} = \boldsymbol{m}_{k,t} \odot \boldsymbol{w}_t$ is the synchronized local weights at the beginning of a communication round.*

**Fact 2** (Local update from client $k$). *The local update of clients can be formalized as follows:*

$$\boldsymbol{U}_{k,t} = \tilde{\boldsymbol{w}}_{k,t,0} - \tilde{\boldsymbol{w}}_{k,t,N} = \eta \sum_{\tau} \boldsymbol{m}_{k,t} \odot \boldsymbol{g}_{k,t,\tau}(\tilde{\boldsymbol{w}}_{k,t,\tau}). \tag{7}$$

**Fact 3** (Server's update). *The server aggregates the sparse update by averaging, which can be formalized as follows:*

$$\boldsymbol{w}_{t+1} = \boldsymbol{w}_t - \frac{1}{S} \sum_{k \in S_t} \boldsymbol{U}_{k,t} = \boldsymbol{w}_t - \frac{\eta}{S} \sum_{k \in S_t, \tau} \boldsymbol{m}_{k,t} \odot \boldsymbol{g}_{k,t,\tau}(\tilde{\boldsymbol{w}}_{k,t,\tau}). \tag{8}$$

**Fact 4** (Global and local loss). *The local loss of a client is denoted by $F_k(\tilde{\boldsymbol{w}}_k)$, and is formulated as:*

$$F_k(\tilde{\boldsymbol{w}}_k) = \mathbb{E}[\mathcal{L}_{(\boldsymbol{x},y) \sim \mathcal{D}_k}(\tilde{\boldsymbol{w}}_k; (\boldsymbol{x}, y))] \tag{9}$$

*where $\tilde{\boldsymbol{w}}_k = \boldsymbol{m}_k^* \odot \boldsymbol{w}$, and the global loss can be formalized as follows:*

$$f(\boldsymbol{w}) = \frac{1}{K} \sum_{k=1}^{K} F_k(\tilde{\boldsymbol{w}}_k) = \frac{1}{K} \sum_{k=1}^{K} F_k(\boldsymbol{m}_k^* \odot \boldsymbol{w}) \tag{10}$$

*and $\{\boldsymbol{m}_k^*\}$ in the function are viewed as optimal personalized masks which satisfies the sparse volume constraint $\|\boldsymbol{1} - \boldsymbol{m}_k^*\|_0 = \beta$ for all $k$.*

## C.2 AUXILIARY LEMMAS

In the following. we shall present several common lemmas that are heavily used in the FL literature.

**Lemma 1** (Cauchy-Schwarz). *Assume arbitrary vector sequences $\{\boldsymbol{a}_k\}_{k=1,\ldots,K}$ and $\{\boldsymbol{b}_k\}_{k=1,\ldots,K}$, Cauchy-Schwarz inequality implies:*

$$\left\| \sum_{k=1}^{K} \boldsymbol{a}_k \boldsymbol{b}_k \right\|^2 \leq \left( \sum_{k=1}^{K} \|\boldsymbol{a}_k\|^2 \right) \left( \sum_{k=1}^{K} \|\boldsymbol{b}_k\|^2 \right), \tag{11}$$

*by taking $b_k = 1$, we also have:*

$$\left\| \sum_{k=1}^{K} \boldsymbol{a}_k \right\|^2 \leq K \left( \sum_{k=1}^{K} \|\boldsymbol{a}_k\|^2 \right), \tag{12}$$

**Lemma 2** (Separating mean and variance, Lemma B.3 (Xu et al., 2021)). *Let $\{\boldsymbol{a}_1, \ldots, \boldsymbol{a}_\tau\}$ be $\tau$ random vectors in $\mathbb{R}^d$. Suppose that $\{\boldsymbol{a}_i - \boldsymbol{\xi}_i\}$ form a martingale difference sequence, i.e. $\mathbb{E}[\boldsymbol{a}_i - \boldsymbol{\xi}_i \mid \boldsymbol{a}_1, \ldots, \boldsymbol{a}_{i-1}] = 0$, and suppose that their variance is bounded by $\mathbb{E}\left[\|\boldsymbol{a}_i - \boldsymbol{\xi}_i\|^2\right] \leq \sigma^2$. Then, the following inequality holds:*

$$\mathbb{E}\left[ \left\| \sum_{i=1}^{\tau} \boldsymbol{a}_i \right\|^2 \right] \leq 2 \left\| \sum_{i=1}^{\tau} \boldsymbol{\xi}_i \right\|^2 + 2\tau\sigma^2.$$

**Lemma 3** (Relaxed triangle inequality, Lemma 3 (Karimireddy et al., 2020)). *Let $\boldsymbol{v}_i$ and $\boldsymbol{v}_j$ be vectors in $\mathbb{R}^d$. Then the following inequality holds true for any $a > 0$:*

$$\|\boldsymbol{v}_i + \boldsymbol{v}_j\|^2 \leq (1 + a) \|\boldsymbol{v}_i\|^2 + \left(1 + \frac{1}{a}\right) \|\boldsymbol{v}_j\|^2. \tag{13}$$

**Lemma 4.** *For random vector $\boldsymbol{v}_1$ satisfying $\mathbb{E}[\boldsymbol{v}_1] = \begin{bmatrix} 0 \\ \cdot \\ \cdot \\ 0 \end{bmatrix}$, and assume another random vector $\boldsymbol{v}_2$ is independent with $\boldsymbol{v}_1$, we have:*

$$\mathbb{E}[\|\boldsymbol{v}_1 + \boldsymbol{v}_2\|^2] = \mathbb{E}[\|\boldsymbol{v}_1\|^2] + \mathbb{E}\|\boldsymbol{v}_2\|^2 \tag{14}$$

*Proof.*

$$
\begin{aligned}
\mathbb{E}[\|\boldsymbol{v}_1 + \boldsymbol{v}_2\|^2] &= \mathbb{E}\langle \boldsymbol{v}_1 + \boldsymbol{v}_2, \boldsymbol{v}_1 + \boldsymbol{v}_2 \rangle \\
&= \mathbb{E}\|\boldsymbol{v}_1\|^2 + \mathbb{E}\|\boldsymbol{v}_2\|^2 + 2\mathbb{E}\langle \boldsymbol{v}_1, \boldsymbol{v}_2 \rangle \\
&= \mathbb{E}\|\boldsymbol{v}_1\|^2 + \mathbb{E}\|\boldsymbol{v}_2\|^2 + 2\langle \mathbb{E}\boldsymbol{v}_1, \mathbb{E}\boldsymbol{v}_2 \rangle \\
&= \mathbb{E}\|\boldsymbol{v}_1\|^2 + \mathbb{E}\|\boldsymbol{v}_2\|^2
\end{aligned}
\tag{15}
$$

This completes the proof. □

### C.3 KEY LEMMAS

In this section, we present several important lemmas that would be used in our formal proof. All the presented claims are rigorously proved.

**Lemma 5** (Smoothness of $f(\boldsymbol{w})$). *Assume $\tilde{\boldsymbol{w}}_k = \boldsymbol{m}_k^* \odot \boldsymbol{w}$ for any $\boldsymbol{m}_k^* \in \{0, 1\}^d$, we have $L$-smoothness for $f(\boldsymbol{w}) = \frac{1}{K} \sum_k F_k(\tilde{\boldsymbol{w}}_k)$, i.e., for any $\boldsymbol{w}_1, \boldsymbol{w}_2 \in \mathbb{R}^d$, we have:*

$$\|\nabla f(\boldsymbol{w}_1) - \nabla f(\boldsymbol{w}_2)\| \leq L\|\boldsymbol{w}_1 - \boldsymbol{w}_2\|. \tag{16}$$

*Proof.*

$$
\begin{aligned}
\|\nabla f(\boldsymbol{w}_1) - \nabla f(\boldsymbol{w}_2)\| &= \left\| \frac{1}{K} \sum_k (\boldsymbol{m}_k^* \odot \nabla F_k(\boldsymbol{m}_k^* \odot \boldsymbol{w}_1) - \boldsymbol{m}_k^* \odot \nabla F_k(\boldsymbol{m}_k^* \odot \boldsymbol{w}_2)) \right\| \\
&\leq \frac{1}{K} \sum_k \|\boldsymbol{m}_k^* \odot (\nabla F_k(\boldsymbol{m}_k^* \odot \boldsymbol{w}_1) - \nabla F_k(\boldsymbol{m}_k^* \odot \boldsymbol{w}_2))\| \\
&\leq \frac{1}{K} \sum_k \|\nabla F_k(\boldsymbol{m}_k^* \odot \boldsymbol{w}_1) - \nabla F_k(\boldsymbol{m}_k^* \odot \boldsymbol{w}_2)\| \\
&\overset{(a)}{\leq} L \frac{1}{K} \sum_k \|\boldsymbol{m}_k^* \odot (\boldsymbol{w}_1 - \boldsymbol{w}_2)\| \\
&\leq L\|\boldsymbol{w}_1 - \boldsymbol{w}_2\|
\end{aligned}
\tag{17}
$$

where the first equality holds by the Fact 4 and inequality (a) is due to Assumption 3. This completes the proof. □

**Lemma 6** (Separating mean and variance of stochastic gradient). *For $\boldsymbol{m}_{k,t} \in \{0, 1\}^d$, We have the following bounding for the expected average gradient:*

$$\mathbb{E}_t \left[ \left\| \frac{1}{S} \sum_{k \in S_t, \tau} \boldsymbol{m}_{k,t} \odot \boldsymbol{g}_{k,t,\tau}(\tilde{\boldsymbol{w}}_{k,t,\tau}) \right\|^2 \right] \leq 2\mathbb{E}_t \left[ \left\| \frac{1}{S} \sum_{k \in S_t, \tau} \boldsymbol{m}_{k,t} \odot \nabla F_k(\tilde{\boldsymbol{w}}_{k,t,\tau}) \right\|^2 \right] + \frac{2N\sigma^2}{S}. \tag{18}$$

*where $\mathbb{E}_t[\cdot]$ denotes the expectation over all the randomness of round $t$.*

*Proof.* We view $\frac{1}{S}\boldsymbol{m}_{k,t} \odot \boldsymbol{g}_{k,t,\tau}(\tilde{\boldsymbol{w}}_{k,t,\tau})$ for all $k$ and $\tau$ as stochastic vectors. By the unbiasedness of stochastic gradient, we know their variance satisfies:

$$
\begin{aligned}
&||\frac{1}{S}\boldsymbol{m}_{k,t} \odot \boldsymbol{g}_{k,t,\tau}(\tilde{\boldsymbol{w}}_{k,t,\tau}) - \mathbb{E}[\frac{1}{S}\boldsymbol{m}_{k,t} \odot \boldsymbol{g}_{k,t,\tau}(\tilde{\boldsymbol{w}}_{k,t,\tau})||^2] \\
=&||\frac{1}{S}\boldsymbol{m}_{k,t} \odot (\boldsymbol{g}_{k,t,\tau}(\tilde{\boldsymbol{w}}_{k,t,\tau}) - \nabla F_k(\tilde{\boldsymbol{w}}_{k,t,\tau}))||^2 \\
\leq&\frac{1}{S^2}||\boldsymbol{g}_{k,t,\tau}(\tilde{\boldsymbol{w}}_{k,t,\tau}) - \nabla F_k(\tilde{\boldsymbol{w}}_{k,t,\tau})||^2 \\
\leq&\frac{\sigma^2}{S^2}
\end{aligned}
\tag{19}
$$

where the first inequality holds since $\boldsymbol{m}_{k,t} \in \{0,1\}^d$ and the last inequality is due to Assumption 2. As per the variance given above, and directly apply Lemma 2, the claim immediately shows. $\square$

**Lemma 7.** *We have the following error bound by introducing personalized masks:*

$$
\|\boldsymbol{m}_{k,t} \odot \nabla F_k(\tilde{\boldsymbol{w}}_{k,t,\tau}) - \boldsymbol{m}_k^* \nabla F_k(\tilde{\boldsymbol{w}}_{k,t})\|^2 \leq \|\nabla F_k(\tilde{\boldsymbol{w}}_{k,t,\tau}) - \nabla F_k(\tilde{\boldsymbol{w}}_{k,t})\|^2 + dist(\boldsymbol{m}_{k,t}, \boldsymbol{m}_k^*)B^2
\tag{20}
$$

*and,*

$$
\|\boldsymbol{m}_{k,t} \odot \nabla F_k(\tilde{\boldsymbol{w}}_{k,t}) - \nabla f(\boldsymbol{w}_t)\|^2 \leq 3(d-\beta)G^2 + \frac{3}{K}\sum_{k'} B^2(dist(\boldsymbol{m}_{k,t}, \boldsymbol{m}_{k',t}) + dist(\boldsymbol{m}_{k',t}, \boldsymbol{m}_{k'}^*))
\tag{21}
$$

*where $dist(\boldsymbol{m}_1, \boldsymbol{m}_2)$ is the hamming distance between two masks.*

*Proof.* Let $\nabla^{(j)} F_k(\tilde{\boldsymbol{w}}_{k,t,\tau})$ and $\nabla^{(j)} F_k(\tilde{\boldsymbol{w}}_{k,t})$ be the derivative over the $j$-th weight coordinates. Define the support of a vector $\boldsymbol{v} \in \mathbb{R}^d$ as $\text{supp}(\boldsymbol{v}) = \{j : \boldsymbol{v}^{(j)} \neq 0\}$. The left hand side of the first claim can be rewritten and bounded as follows:

$$
\begin{aligned}
&\|\boldsymbol{m}_{k,t} \odot \nabla F_k(\tilde{\boldsymbol{w}}_{k,t,\tau}) - \boldsymbol{m}_k^* \odot \nabla F_k(\tilde{\boldsymbol{w}}_{k,t})\|^2 \\
=&\sum_{j \in \text{supp}(\boldsymbol{m}_{k,t}) \cap \text{supp}(\boldsymbol{m}_k^*)} \|\nabla^{(j)} F_k(\tilde{\boldsymbol{w}}_{k,t,\tau}) - \nabla^{(j)} F_k(\tilde{\boldsymbol{w}}_{k,t})\|^2 \\
&+ \sum_{j \in \text{supp}(\boldsymbol{m}_{k,t}) \text{and} j \notin \text{supp}(\boldsymbol{m}_k^*)} \|\nabla^{(j)} F_k(\tilde{\boldsymbol{w}}_{k,t,\tau})\|^2 + \sum_{j \notin \text{supp}(\boldsymbol{m}_{k,t}) \text{and} j \in \text{supp}(\boldsymbol{m}_k^*)} \|\nabla^{(j)} F_k(\tilde{\boldsymbol{w}}_{k,t})\|^2 \\
\overset{(a)}{\leq}&\sum_{j \in \text{supp}(\boldsymbol{m}_{k,t}) \cap \text{supp}(\boldsymbol{m}_k^*)} \|\nabla^{(j)} F_k(\tilde{\boldsymbol{w}}_{k,t,\tau}) - \nabla^{(j)} F_k(\tilde{\boldsymbol{w}}_{k,t})\|^2 + \sum_{j \in \text{supp}(\boldsymbol{m}_{k,t} \oplus \boldsymbol{m}_k^*)} B^2 \\
\overset{(b)}{=}&\sum_{j \in \text{supp}(\boldsymbol{m}_{k,t}) \cap \text{supp}(\boldsymbol{m}_k^*)} \|\nabla^{(j)} F_k(\tilde{\boldsymbol{w}}_{k,t,\tau}) - \nabla^{(j)} F_k(\tilde{\boldsymbol{w}}_{k,t})\|^2 + dist(\boldsymbol{m}_{k,t}, \boldsymbol{m}_k^*)B^2 \\
\leq&\sum_j \|\nabla^{(j)} F_k(\tilde{\boldsymbol{w}}_{k,t,\tau}) - \nabla^{(j)} F_k(\tilde{\boldsymbol{w}}_{k,t})\|^2 + dist(\boldsymbol{m}_{k,t}, \boldsymbol{m}_k^*)B^2 \\
\leq&\|\nabla F_k(\tilde{\boldsymbol{w}}_{k,t,\tau}) - \nabla F_k(\tilde{\boldsymbol{w}}_{k,t})\|^2 + dist(\boldsymbol{m}_{k,t}, \boldsymbol{m}_k^*)B^2
\end{aligned}
\tag{22}
$$

where inequality $(a)$ is true since we have $\|\nabla^{(j)} F_k(\tilde{\boldsymbol{w}}_{k,t,\tau})\|^2 \leq B^2$ and $\|\nabla^{(j)} F_k(\tilde{\boldsymbol{w}}_{k,t})\|^2 \leq B^2$ by coordinate-wise bounded gradient Assumption 4. Inequality (b) is due to the definition of hamming distance, $dist(\boldsymbol{v}_1, \boldsymbol{v}_2) = \sum_{j \in \text{supp}(\boldsymbol{v}_1 \oplus \boldsymbol{v}_2)}$. This shows our first claim.

Following the same technique, we expand the left hand side of the second claim using Cauchy Schwarz inequality in Lemma 1, as follows:

$$\|\boldsymbol{m}_{k,t} \odot \nabla F_k(\tilde{\boldsymbol{w}}_{k,t}) - \nabla f(\boldsymbol{w}_t)\|^2$$

$$=\|\boldsymbol{m}_{k,t} \odot \nabla F_k(\tilde{\boldsymbol{w}}_{k,t}) - \frac{1}{K}\sum_{k'} \boldsymbol{m}_{k'}^* \odot \nabla F_{k'}(\tilde{\boldsymbol{w}}_{k',t})\|^2$$

$$\leq 3\|\boldsymbol{m}_{k,t} \odot \nabla F_k(\tilde{\boldsymbol{w}}_{k,t}) - \frac{1}{K}\sum_{k'} \boldsymbol{m}_{k,t} \odot \nabla F_{k'}(\tilde{\boldsymbol{w}}_{k',t})\|^2$$

$$+ 3\|\frac{1}{K}\sum_{k'} (\boldsymbol{m}_{k,t} \odot \nabla F_{k'}(\tilde{\boldsymbol{w}}_{k',t}) - \boldsymbol{m}_{k',t} \odot \nabla F_{k'}(\tilde{\boldsymbol{w}}_{k',t}))\|^2$$

$$+ 3\|\frac{1}{K}\sum_{k'} (\boldsymbol{m}_{k',t} \odot \nabla F_{k'}(\tilde{\boldsymbol{w}}_{k',t}) - \boldsymbol{m}_{k'}^* \odot \nabla F_{k'}(\tilde{\boldsymbol{w}}_{k',t}))\|^2$$

$$\leq 3\|\boldsymbol{m}_{k,t} \odot \nabla F_k(\tilde{\boldsymbol{w}}_{k,t}) - \frac{1}{K}\sum_{k'} \boldsymbol{m}_{k,t} \odot \nabla F_{k'}(\tilde{\boldsymbol{w}}_{k',t})\|^2 \tag{23}$$

$$+ \frac{3}{K}\sum_{k'} \|\boldsymbol{m}_{k,t} \odot \nabla F_{k'}(\tilde{\boldsymbol{w}}_{k',t}) - \boldsymbol{m}_{k',t} \odot \nabla F_{k'}(\tilde{\boldsymbol{w}}_{k',t})\|^2$$

$$+ \frac{3}{K}\sum_{k'} \|(\boldsymbol{m}_{k',t} \odot \nabla F_{k'}(\tilde{\boldsymbol{w}}_{k',t}) - \boldsymbol{m}_{k'}^* \odot \nabla F_{k'}(\tilde{\boldsymbol{w}}_{k',t}))\|^2$$

$$\leq 3(d-\beta)G^2 + \frac{3}{K}\sum_{k'} dist(\boldsymbol{m}_{k,t}, \boldsymbol{m}_{k',t})B^2 + \frac{3}{K}\sum_{k'} dist(\boldsymbol{m}_{k',t}, \boldsymbol{m}_{k'}^*)B^2$$

$$=3(d-\beta)G^2 + \frac{3}{K}\sum_{k'} B^2(dist(\boldsymbol{m}_{k,t}, \boldsymbol{m}_{k',t}) + dist(\boldsymbol{m}_{k',t}, \boldsymbol{m}_{k'}^*))$$

For the last inequality, we use the same technique as in the first claim to bound the second term and the third term, and the first term is bounded by Lemma 8. $\qquad\square$

**Lemma 8.** *Let $\|\boldsymbol{1} - \boldsymbol{m}\|_0 = \beta$ be the sparse volume of a mask $\boldsymbol{m}$, and suppose $\boldsymbol{m} \in \{0,1\}^d$, by assumption 4, we have the following relation:*

$$\|\boldsymbol{m} \odot (\nabla F_k(\tilde{\boldsymbol{w}}_{k,t}) - \frac{1}{K}\sum_{k'} \nabla F_{k'}(\tilde{\boldsymbol{w}}_{k',t}))\|^2 \leq (d-\beta)G^2 \tag{24}$$

*and,*

$$\|\nabla F_k(\tilde{\boldsymbol{w}}_{k,t}) - \frac{1}{K}\sum_{k'} \boldsymbol{m} \odot \nabla F_{k'}(\tilde{\boldsymbol{w}}_{k',t})\|^2 \leq (d-\beta)G^2 + \beta B^2 \tag{25}$$

*where $d$ is the dimension of the models (or identically, the dimension of the mask $\boldsymbol{m}$).*

*Proof.* The proof of this lemma follows a similar technique in Lemma 7. Let $\nabla^{(j)} F_k(\tilde{\boldsymbol{w}}_{k,t})$ be the derivative over the $j$-th weight coordinate. Define the support of a vector $\boldsymbol{v} \in \mathbb{R}^d$ as $\text{supp}(\boldsymbol{v}) = \{j : \boldsymbol{v}^{(j)} \neq 0\}$. The left hand side (L.H.S) of our first claim can be bounded by:

$$\|\boldsymbol{m} \odot (\nabla F_k(\tilde{\boldsymbol{w}}_{k,t}) - \frac{1}{K}\sum_{k'} \nabla F_{k'}(\tilde{\boldsymbol{w}}_{k',t}))\|^2$$

$$= \sum_{j \in \text{supp}(\boldsymbol{m})} \|\nabla^{(j)} F_k(\tilde{\boldsymbol{w}}_{k,t}) - \frac{1}{K}\sum_{k'} \nabla^{(j)} F_{k'}(\tilde{\boldsymbol{w}}_{k',t})\|^2 \tag{26}$$

$$\leq (d-\beta)G^2$$

where the last inequality is obtained from the sparsity constraint and the coordinate-wise gradient dissimilarity Assumption 1. This shows our first claim.

For our second claim, we expand its L.H.S as follows:

$$\|\nabla F_k(\tilde{\boldsymbol{w}}_{k,t}) - \frac{1}{K}\sum_{k'} \boldsymbol{m} \odot \nabla F_{k'}(\tilde{\boldsymbol{w}}_{k',t})\|^2$$

$$= \sum_{j\in\mathrm{supp}(\boldsymbol{m})} \|\nabla^{(j)} F_k(\tilde{\boldsymbol{w}}_{k,t}) - \frac{1}{K}\sum_{k'} \nabla^{(j)} F_{k'}(\tilde{\boldsymbol{w}}_{k',t})\|^2 + \sum_{j\notin\mathrm{supp}(\boldsymbol{m})} \|\nabla^{(j)} F_k(\tilde{\boldsymbol{w}}_{k,t})\|^2 \quad (27)$$

$$\leq (d-\beta)G^2 + \beta B^2,$$

where the last inequality is due to Assumption 1 and Assumption 4. It completes the proof. $\qquad\square$

**Lemma 9** (Drift towards Synchronized Point). *For any* $t \in \{1,\ldots,T\}$, $\tau \in \{0,\ldots,N\}$, *and learning rate satisfies* $\eta \leq \frac{1}{16LN}$, *we have the following claim:*

$$\frac{1}{K}\sum_k \mathbb{E}_t\left[\|\tilde{\boldsymbol{w}}_{k,t,\tau} - \tilde{\boldsymbol{w}}_{k,t}\|^2\right] \leq 5N\eta^2(\sigma^2 + 18N\Phi_t) + 30N^2\eta^2\|\nabla f(\boldsymbol{w}_t)\|^2 \quad (28)$$

*where* $\Phi_t = \frac{1}{K}\sum_k((d-\beta)G^2 + \frac{1}{K}\sum_{k'} B^2(dist(\boldsymbol{m}_{k,t},\boldsymbol{m}_{k',t}) + dist(\boldsymbol{m}_{k',t},\boldsymbol{m}_{k'}^*)))$ *and* $\mathbb{E}_t[\cdot]$ *denotes the expectation over all the randomness of round* $t$.

*Proof.* We follow the basic techniques from (Reddi et al., 2020) to prove this lemma.

We first assume that:

- $O = \boldsymbol{m}_{k,t} \odot \boldsymbol{g}_{k,t,\tau-1}(\tilde{\boldsymbol{w}}_{k,t,\tau-1}) - \boldsymbol{m}_{k,t} \odot \nabla F_k(\tilde{\boldsymbol{w}}_{k,t,\tau-1})$
- $P = \boldsymbol{m}_{k,t} \odot \nabla F_k(\tilde{\boldsymbol{w}}_{k,t,\tau-1}) - \boldsymbol{m}_{k,t} \odot \nabla F_k(\tilde{\boldsymbol{w}}_{k,t})$
- $Q = \boldsymbol{m}_{k,t} \odot \nabla F_k(\tilde{\boldsymbol{w}}_{k,t}) - \nabla f(\boldsymbol{w}_t)$

We can expand $\tilde{\boldsymbol{w}}_{k,t,\tau}$ as follows:

$$\frac{1}{K}\sum_k \mathbb{E}_t\left[\|\tilde{\boldsymbol{w}}_{k,t,\tau} - \tilde{\boldsymbol{w}}_t\|^2\right]$$

$$\overset{\text{Fact 1}}{=} \frac{1}{K}\sum_k \mathbb{E}_t\left[\|\tilde{\boldsymbol{w}}_{k,t,\tau-1} - \tilde{\boldsymbol{w}}_t - \eta\boldsymbol{m}_{k,t} \odot \boldsymbol{g}_{k,t,\tau-1}(\tilde{\boldsymbol{w}}_{k,t,\tau-1})\|^2\right]$$

$$= \frac{1}{K}\sum_k \mathbb{E}_t\left[\|\tilde{\boldsymbol{w}}_{k,t,\tau-1} - \tilde{\boldsymbol{w}}_t - \eta(O + P + Q + \nabla f(\boldsymbol{w}_t))\|^2\right] \quad (29)$$

$$\leq \frac{1 + \frac{1}{2N-1}}{K}\sum_k \mathbb{E}_t\|\tilde{\boldsymbol{w}}_{k,t,\tau-1} - \tilde{\boldsymbol{w}}_t\|^2 + \frac{\eta^2}{K}\sum_k \mathbb{E}_t\|O\|^2 + \frac{6N\eta^2}{K}\sum_k \mathbb{E}_t\|P\|^2$$

$$+ \frac{6N\eta^2}{K}\sum_k \mathbb{E}_t\|Q\|^2 + \frac{6N\eta^2}{K}\sum_k \|\nabla f(\boldsymbol{w}_t)\|^2$$

where the last inequality follows from Lemma 3 and Lemma 4. Explicitly, we use Lemma 4 to treat the stochastic term with $\mathbb{E}_t\|O\|^2$, and then we use Lemma 3 with $a = 2N-1$ to separate the other four terms.

Then we proceed by separately bounding the components in the above inequality.

**Bounding the second term:**

$$\frac{\eta^2}{K}\sum_k \mathbb{E}_t\|O\|^2 = \frac{\eta^2}{K}\sum_k \mathbb{E}_t\|\boldsymbol{m}_{k,t} \odot \boldsymbol{g}_{k,t,\tau-1}(\tilde{\boldsymbol{w}}_{k,t,\tau-1}) - \boldsymbol{m}_{k,t} \odot \nabla F_k(\tilde{\boldsymbol{w}}_{k,t,\tau-1})\|^2$$

$$= \frac{\eta^2}{K}\sum_k \mathbb{E}_t\|\boldsymbol{m}_{k,t} \odot (\boldsymbol{g}_{k,t,\tau-1}(\tilde{\boldsymbol{w}}_{k,t,\tau-1}) - \nabla F_k(\tilde{\boldsymbol{w}}_{k,t,\tau-1}))\|^2 \quad (30)$$

$$\leq \frac{\eta^2}{K}\sum_k \mathbb{E}_t\|\boldsymbol{g}_{k,t,\tau-1}(\tilde{\boldsymbol{w}}_{k,t,\tau-1}) - \nabla F_k(\tilde{\boldsymbol{w}}_{k,t,\tau-1})\|^2$$

$$\leq \eta^2\sigma^2$$

where the last inequality holds by Assumption 2.

**Bounding the third term:**

$$
\frac{6N\eta^2}{K}\sum_k \mathbb{E}_t\|P\|^2 = \frac{6N\eta^2}{K}\sum_k \mathbb{E}_t\|\boldsymbol{m}_{k,t}\odot\nabla F_k(\tilde{\boldsymbol{w}}_{k,t,\tau-1}) - \boldsymbol{m}_{k,t}\odot\nabla F_k(\tilde{\boldsymbol{w}}_{k,t})\|^2
$$

$$
\leq \frac{6N\eta^2}{K}\sum_k \mathbb{E}_t\|\nabla F_k(\tilde{\boldsymbol{w}}_{k,t,\tau-1}) - \nabla F_k(\tilde{\boldsymbol{w}}_{k,t})\|^2 \tag{31}
$$

$$
\leq (6N\eta^2 L^2)\frac{1}{K}\sum_k \mathbb{E}_t\|\tilde{\boldsymbol{w}}_{k,t,\tau-1} - \tilde{\boldsymbol{w}}_{k,t}\|^2
$$

**Bounding the fourth term:**

$$
\frac{6N\eta^2}{K}\sum_k \mathbb{E}_t\|Q\|^2
$$

$$
= \frac{6N\eta^2}{K}\sum_k \mathbb{E}_t\|\boldsymbol{m}_{k,t}\odot\nabla F_k(\tilde{\boldsymbol{w}}_{k,t}) - \nabla f(\boldsymbol{w}_t)\|^2 \tag{32}
$$

$$
\leq \frac{18N\eta^2}{K}\sum_k((d-\beta)G^2 + \frac{1}{K}\sum_{k'}B^2(dist(\boldsymbol{m}_{k,t},\boldsymbol{m}_{k',t}) + dist(\boldsymbol{m}_{k',t},\boldsymbol{m}_{k'}^*))),
$$

where we use the second claim in Lemma 7 in the last inequality.

Let $\Phi_t = \frac{1}{K}\sum_k((d-\beta)G^2 + \frac{1}{K}\sum_{k'}B^2(dist(\boldsymbol{m}_{k,t},\boldsymbol{m}_{k',t}) + dist(\boldsymbol{m}_{k',t},\boldsymbol{m}_{k'}^*)))$. The bound can be simplified as follows:

$$
\frac{6N\eta^2}{K}\sum_k \mathbb{E}_t\|Q\|^2 \leq 18N\eta^2\Phi_t. \tag{33}
$$

**Putting together:** Plugging all the components into Eq.(29) , the following result immediately follows:

$$
\frac{1}{K}\sum_k \mathbb{E}_t\left[\|\tilde{\boldsymbol{w}}_{k,t,\tau} - \tilde{\boldsymbol{w}}_{k,t}\|^2\right]
$$

$$
\leq (1 + \frac{1}{2N-1} + 6N\eta^2 L^2)\frac{1}{K}\sum_k \mathbb{E}_t\|\tilde{\boldsymbol{w}}_{k,t,\tau-1} - \tilde{\boldsymbol{w}}_{k,t}\|^2 + \eta^2\sigma^2 + 18N\eta^2\Phi_t + 6N\eta^2\|\nabla f(\boldsymbol{w}_t)\|^2
$$

$$
\leq (1 + \frac{1}{N-1})\frac{1}{K}\sum_k \mathbb{E}_t\|\tilde{\boldsymbol{w}}_{k,t,\tau-1} - \tilde{\boldsymbol{w}}_{k,t}\|^2 + \eta^2(\sigma^2 + 18N\Phi_t) + 6N\eta^2\|\nabla f(\boldsymbol{w}_t)\|^2,
$$

$$
\tag{34}
$$

where the last inequlity holds by our assumption $\eta \leq \frac{1}{16LN}$.

**Unrolling the recursion,** we obtain the following results:

$$
\frac{1}{K}\sum_k \mathbb{E}_t\left[\|\tilde{\boldsymbol{w}}_{k,t,\tau} - \tilde{\boldsymbol{w}}_{k,t}\|^2\right]
$$

$$
\leq \sum_{\tau=0}^{N-1}(1 + \frac{1}{N-1})^\tau \left[\eta^2(\sigma^2 + 18N\Phi_t) + 6N\eta^2\|\nabla f(\boldsymbol{w}_t)\|^2\right] \tag{35}
$$

$$
\leq (N-1)\times\left((1 + \frac{1}{N-1})^N - 1\right)\left[\eta^2(\sigma^2 + 18N\Phi_t) + 6N\eta^2\|\nabla f(\boldsymbol{w}_t)\|^2\right]
$$

$$
\leq 5N\eta^2(\sigma^2 + 18N\Phi_t) + 30N^2\eta^2\|\nabla f(\boldsymbol{w}_t)\|^2
$$

The last inequality holds since $\left((1 + \frac{1}{N-1})^N - 1\right) \leq 5$ for $N \geq 1$. This completes the proof.  □

## C.4 FORMAL PROOF

We start our proof by expanding $f(\boldsymbol{w}_{t+1})$ under its smoothness condition (see Lemma 5), which indicates that:

$$
\begin{aligned}
&\mathbb{E}_t\left[f\left(\boldsymbol{w}_{t+1}\right) \mid \boldsymbol{w}_t\right] \\
&\leq f(\boldsymbol{w}_t) - \langle \nabla f(\boldsymbol{w}_t), \mathbb{E}_t[\boldsymbol{w}_{t+1} - \boldsymbol{w}_t]\rangle + \frac{L}{2}\mathbb{E}_t\|\boldsymbol{w}_{t+1} - \boldsymbol{w}_t\|^2 \\
&\overset{\text{Fact } 3}{=} f\left(\boldsymbol{w}_t\right) - \eta\mathbb{E}_t\left[\left\langle\nabla f\left(\boldsymbol{w}_t\right), \frac{1}{S}\sum_{k\in S_t}\boldsymbol{U}_{k,t}\right\rangle\right] + \frac{L}{2}\mathbb{E}_t\left\|\frac{1}{S}\sum_{k\in S_t}\boldsymbol{U}_{k,t}\right\|^2 \\
&\overset{\text{Fact } 2}{=} f\left(\boldsymbol{w}_t\right) - \frac{\eta}{N}\left\langle N\nabla f\left(\boldsymbol{w}_t\right), \mathbb{E}_t\left[\frac{1}{K}\sum_{k,\tau}\boldsymbol{m}_{k,t}\odot\nabla F_k(\tilde{\boldsymbol{w}}_{k,t,\tau})\right]\right\rangle + \frac{L}{2}\mathbb{E}_t\left\|\frac{1}{S}\sum_{k\in S_t}\boldsymbol{U}_{k,t}\right\|^2 \\
&\leq f\left(\boldsymbol{w}_t\right) - \frac{\eta N}{2}\|\nabla f\left(\boldsymbol{w}_t\right)\|^2 + \underbrace{\frac{\eta}{2N}\mathbb{E}_t\|\frac{1}{K}\sum_{k,\tau}\boldsymbol{m}_{k,t}\odot\nabla F_k(\tilde{\boldsymbol{w}}_{k,t,\tau}) - N\nabla f\left(\boldsymbol{w}_t\right)\|^2}_{T_1} \\
&\qquad\qquad\qquad\qquad\qquad\qquad\qquad\qquad + \underbrace{\frac{L}{2}\mathbb{E}_t\left\|\frac{1}{S}\sum_{k\in S_t}\boldsymbol{U}_{k,t}\right\|^2}_{T_2}
\end{aligned}
\tag{36}
$$

where the last inequality holds since $-ab \leq \frac{1}{2}((b-a)^2 - a^2)$, and $\mathbb{E}_t[\cdot]$ is the expectation over all the randomness in round $t$.

In the following, we shall separately bound $T_1$ and $T_2$.

**Bounding $T_1$:**

$$
\begin{aligned}
T_1 &= \frac{\eta}{2N}\mathbb{E}_t\|\frac{1}{K}\sum_{k,\tau}\boldsymbol{m}_{k,t}\odot\nabla F_k(\tilde{\boldsymbol{w}}_{k,t,\tau}) - N\nabla f\left(\boldsymbol{w}_t\right)\|^2 \\
&\overset{\text{Fact } 4}{=} \frac{\eta}{2N}\mathbb{E}_t\|\frac{1}{K}\sum_{k,\tau}\boldsymbol{m}_{k,t}\odot\nabla F_k(\tilde{\boldsymbol{w}}_{k,t,\tau}) - N\frac{1}{K}\sum_k\boldsymbol{m}_k^*\odot\nabla F_k(\tilde{\boldsymbol{w}}_{k,t})\|^2 \\
&= \frac{\eta}{2N}\mathbb{E}_t\|\frac{1}{K}\sum_{k,\tau}(\boldsymbol{m}_{k,t}\odot\nabla F_k(\tilde{\boldsymbol{w}}_{k,t,\tau}) - \boldsymbol{m}_k^*\odot\nabla F_k(\tilde{\boldsymbol{w}}_{k,t}))\|^2 \\
&\overset{(a)}{\leq} \frac{\eta}{2K}\sum_{k,\tau}\mathbb{E}_t\|\boldsymbol{m}_{k,t}\odot\nabla F_k(\tilde{\boldsymbol{w}}_{k,t,\tau}) - \boldsymbol{m}_k^*\odot\nabla F_k(\tilde{\boldsymbol{w}}_{k,t})\|^2 \\
&\overset{(b)}{\leq} \frac{\eta}{2K}\sum_{k,\tau}(\|\nabla F_k(\tilde{\boldsymbol{w}}_{k,t,\tau}) - \nabla F_k(\tilde{\boldsymbol{w}}_{k,t})\|^2 + dist(\boldsymbol{m}_{k,t}, \boldsymbol{m}_k^*)B^2) \\
&\leq \frac{\eta L^2}{2K}\sum_{k,\tau}\mathbb{E}_t\|\tilde{\boldsymbol{w}}_{k,t,\tau} - \tilde{\boldsymbol{w}}_{k,t}\|^2 + \frac{\eta N}{2K}\sum_k dist(\boldsymbol{m}_{k,t}, \boldsymbol{m}_k^*)B^2
\end{aligned}
\tag{37}
$$

where inequality (a) is due to Cauchy-Schwarz inequality (i.e., Lemma 1), (b) follows from the first claim in Lemma 7, and the last inequality holds by Assumption 3.

Plugging the results of Lemma 9, we obtain that:

$$
\begin{aligned}
T_1 &\leq \frac{\eta L^2 N}{2}(5N\eta^2(\sigma^2 + 18N\Phi_t) + 30N^2\eta^2\|\nabla f(\boldsymbol{w}_t)\|^2) + \frac{\eta N}{2K}\sum_k dist(\boldsymbol{m}_{k,t}, \boldsymbol{m}_k^*)B^2 \\
&\leq \frac{5N^2\eta^3 L^2}{2}(\sigma^2 + 18N\Phi_t) + 15N^3\eta^3 L^2\|\nabla f(\boldsymbol{w}_t)\|^2 + \frac{\eta N}{2K}\sum_k dist(\boldsymbol{m}_{k,t}, \boldsymbol{m}_k^*)B^2
\end{aligned}
\tag{38}
$$

**Bounding $T_2$:**

$$
\begin{aligned}
T_2 =& \frac{L}{2}\mathbb{E}_t \left\| \frac{1}{S}\sum_{k\in S_t} \boldsymbol{U}_{k,t} \right\|^2 \\
\stackrel{\text{Fact 2}}{=}& \frac{L\eta^2}{2}\mathbb{E}_t \left[ \left\| \frac{1}{S}\sum_{k\in S_t,\tau} \boldsymbol{m}_{k,t}\odot\boldsymbol{g}_{k,t,\tau}(\tilde{\boldsymbol{w}}_{k,t,\tau}) \right\|^2 \right] \\
\stackrel{(a)}{\leq}& L\eta^2\mathbb{E}_t \left[ \left\| \frac{1}{S}\sum_{k\in S_t,\tau} \boldsymbol{m}_{k,t}\odot\nabla F_k(\tilde{\boldsymbol{w}}_{k,t,\tau}) \right\|^2 \right] + \frac{NL\eta^2\sigma^2}{S} \\
\leq& L\eta^2 N \sum_\tau \underbrace{\mathbb{E}_t \left[ \left\| \frac{1}{S}\sum_{k\in S_t} \boldsymbol{m}_{k,t}\odot\nabla F_k(\tilde{\boldsymbol{w}}_{k,t,\tau}) \right\|^2 \right]}_{T_3} + \frac{NL\eta^2\sigma^2}{S}
\end{aligned}
\tag{39}
$$

where (a) is obtained as per Lemma 6.

**Bounding $T_3$:**

$$
\begin{aligned}
&T_3 \\
=& \frac{1}{S^2}\mathbb{E}_t \left\langle \sum_{i\in[K]}\mathbb{I}_{\{i\in S_t\}}\boldsymbol{m}_{i,t}\odot\nabla F_i(\tilde{\boldsymbol{w}}_{i,t,\tau}), \sum_{j\in[K]}\mathbb{I}_{\{j\in S_t\}}\boldsymbol{m}_{j,t}\odot\nabla F_j(\tilde{\boldsymbol{w}}_{j,t,\tau}) \right\rangle \\
=& \frac{1}{S^2}\mathbb{E}_t \Bigg[ \sum_{i,j\in[K],j\neq i,\tau} \mathbb{E}_{S_t}[\mathbb{I}_{\{i\in S_t\cap j\in S_t\}}] \langle \boldsymbol{m}_{i,t}\odot\nabla F_i(\tilde{\boldsymbol{w}}_{i,t,\tau}), \boldsymbol{m}_{j,t}\odot\nabla F_j(\tilde{\boldsymbol{w}}_{j,t,\tau}) \rangle \\
&+ \sum_i \mathbb{E}_{S_t}[\mathbb{I}_{\{i\in S_t\}}]\|\boldsymbol{m}_{i,t}\odot\nabla F_i(\tilde{\boldsymbol{w}}_{i,t,\tau})\|^2 \Bigg] \\
=& \frac{1}{S^2}\mathbb{E}_t \Bigg[ \sum_{i,j\in[K],j\neq i} \frac{S(S-1)}{K(K-1)}\langle \boldsymbol{m}_{i,t}\odot\nabla F_i(\tilde{\boldsymbol{w}}_{i,t,\tau}), \boldsymbol{m}_{j,t}\odot\nabla F_j(\tilde{\boldsymbol{w}}_{j,t,\tau}) \rangle + \sum_i \frac{S}{K}\|\boldsymbol{m}_{i,t}\odot\nabla F_i(\tilde{\boldsymbol{w}}_{i,t,\tau})\|^2 \Bigg] \\
=& \frac{1}{S^2}\mathbb{E}_t \Bigg[ \sum_{i,j\in[K]} \frac{S(S-1)}{K(K-1)}\langle \boldsymbol{m}_{i,t}\odot\nabla F_i(\tilde{\boldsymbol{w}}_{i,t,\tau}), \boldsymbol{m}_{j,t}\odot\nabla F_j(\tilde{\boldsymbol{w}}_{j,t,\tau}) \rangle \\
&+ \sum_i \frac{S(K-S)}{K(K-1)}\|\boldsymbol{m}_{i,t}\odot\nabla F_i(\tilde{\boldsymbol{w}}_{i,t,\tau})\|^2 \Bigg] \\
\leq& \mathbb{E}_t \Bigg[ \underbrace{\frac{1}{K^2}\|\sum_k \boldsymbol{m}_{k,t}\odot\nabla F_k(\tilde{\boldsymbol{w}}_{k,t,\tau})\|^2}_{T_4} + \underbrace{\frac{(K-S)}{SK(K-1)}\sum_k \|\boldsymbol{m}_{k,t}\odot\nabla F_k(\tilde{\boldsymbol{w}}_{k,t,\tau})\|^2}_{T_5} \Bigg]
\end{aligned}
\tag{40}
$$

where the last inequality holds since $\frac{S-1}{SK(K-1)} = \frac{S-1}{SK^2-SK} \leq \frac{S}{SK^2} = \frac{1}{K^2}$.

**Bounding $T_4$:**

$$T_4 = \left\| \left( \frac{1}{K} \sum_k \boldsymbol{m}_{k,t} \odot \nabla F_k(\tilde{\boldsymbol{w}}_{k,t,\tau}) - \nabla f(\boldsymbol{w}_t) \right) + \nabla f(\boldsymbol{w}_t) \right\|^2$$

$$\overset{\text{Lemma 1}}{\leq} 2 \left\| \frac{1}{K} \sum_k \boldsymbol{m}_{k,t} \odot \nabla F_k(\tilde{\boldsymbol{w}}_{k,t,\tau}) - \nabla f(\boldsymbol{w}_t) \right\|^2 + 2 \|\nabla f(\boldsymbol{w}_t)\|^2$$

$$\overset{\text{Fact 2}}{=} 2 \left\| \frac{1}{K} \sum_k (\boldsymbol{m}_{k,t} \odot \nabla F_k(\tilde{\boldsymbol{w}}_{k,t,\tau}) - \boldsymbol{m}_k^* \odot \nabla F_k(\tilde{\boldsymbol{w}}_{k,t})) \right\|^2 + 2 \|\nabla f(\boldsymbol{w}_t)\|^2 \tag{41}$$

$$\overset{\text{Lemma 1}}{\leq} \frac{2}{K} \sum_k \|\boldsymbol{m}_{k,t} \odot \nabla F_k(\tilde{\boldsymbol{w}}_{k,t,\tau}) - \boldsymbol{m}_k^* \boldsymbol{\nabla} F_k(\tilde{\boldsymbol{w}}_{k,t})\|^2 + 2 \|\nabla f(\boldsymbol{w}_t)\|^2$$

$$\overset{\text{Lemma 7}}{\leq} \frac{2}{K} \sum_k (\|\nabla F_k(\tilde{\boldsymbol{w}}_{k,t,\tau}) - \nabla F_k(\tilde{\boldsymbol{w}}_{k,t})\|^2 + dist(\boldsymbol{m}_{k,t}, \boldsymbol{m}_k^*)B^2) + 2 \|\nabla f(\boldsymbol{w}_t)\|^2$$

$$\leq \frac{2L^2}{K} \sum_k \|\tilde{\boldsymbol{w}}_{k,t,\tau} - \tilde{\boldsymbol{w}}_{k,t}\|^2 + \frac{2}{K} \sum_k dist(\boldsymbol{m}_{k,t}, \boldsymbol{m}_k^*)B^2 + 2 \|\nabla f(\boldsymbol{w}_t)\|^2,$$

where the last inequality is obtained by the L-smoothness Assumption 3.

**Bounding $T_5$:**

$$T_5 = \frac{(K-S)}{SK(K-1)} \sum_k \|\boldsymbol{m}_{k,t} \odot \nabla F_k(\tilde{\boldsymbol{w}}_{k,t,\tau})\|^2$$

$$\leq \frac{3(K-S)}{SK(K-1)} \sum_k (\|\boldsymbol{m}_{k,t} \odot \nabla F_k(\tilde{\boldsymbol{w}}_{k,t,\tau}) - \boldsymbol{m}_{k,t} \odot \nabla F_k(\tilde{\boldsymbol{w}}_{k,t})\|^2$$

$$+ \|\boldsymbol{m}_{k,t} \odot \nabla F_k(\tilde{\boldsymbol{w}}_{k,t}) - \nabla f(\boldsymbol{w}_t)\|^2 + \|\nabla f(\boldsymbol{w}_t)\|^2)$$

$$\overset{(a)}{\leq} \frac{3L^2(K-S)}{SK(K-1)} \sum_k \|\tilde{\boldsymbol{w}}_{k,t,\tau} - \tilde{\boldsymbol{w}}_{k,t}\|^2 + \frac{3(K-S)}{S(K-1)K} \sum_k \left( 3(d-\beta)G^2 \right. \tag{42}$$

$$+ \frac{3}{K} \sum_{k'} B^2 (dist(\boldsymbol{m}_{k,t}, \boldsymbol{m}_{k',t}) + dist(\boldsymbol{m}_{k',t}, \boldsymbol{m}_{k'}^*))) + \frac{3(K-S)}{S(K-1)} \|\nabla f(\boldsymbol{w}_t)\|^2$$

$$\leq \frac{3L^2}{K} \sum_k \|\tilde{\boldsymbol{w}}_{k,t,\tau} - \tilde{\boldsymbol{w}}_{k,t}\|^2 + 9\Phi_t + 3\|\nabla f(\boldsymbol{w}_t)\|^2,$$

where the last inequality holds since $\frac{K-S}{S(K-1)} \leq 1$ under the condition $S \geq 1$. Inequality (a) is obtained by Assumption 3 and the second claim in Lemma 7 .

**Summing $T_4$ and $T_5$,** we have the following bounding for $T_3$:

$$T_3 \leq \frac{5L^2}{K} \sum_k \|\tilde{\boldsymbol{w}}_{k,t,\tau} - \tilde{\boldsymbol{w}}_{k,t}\| + \frac{2}{K} \sum_k dist(\boldsymbol{m}_{k,t}, \boldsymbol{m}_k^*)B^2 + 9\Phi_t + 5\|\nabla f(\boldsymbol{w}_t)\|^2 \tag{43}$$

Plugging Lemma 9 into the above inequality, we have:

$$T_3$$

$$\leq 5L^2 \left( 5N\eta^2(\sigma^2 + 18N\Phi_t) + 30N^2\eta^2 \|\nabla f(\boldsymbol{w}_t)\|^2 \right) + \frac{2}{K} \sum_k dist(\boldsymbol{m}_{k,t}, \boldsymbol{m}_k^*)B^2 + 9\Phi_t + 5\|\nabla f(\boldsymbol{w}_t)\|^2$$

$$= 25\eta^2 L^2 N(\sigma^2 + 18N\Phi_t) + (150N^2\eta^2 L^2 + 5)\|\nabla f(\boldsymbol{w}_t)\|^2 + \frac{2}{K} \sum_k dist(\boldsymbol{m}_{k,t}, \boldsymbol{m}_k^*)B^2 + 9\Phi_t \tag{44}$$

**Plugging $T_3$ into Inequality (39),** we bound $T_2$ as follows:

$$T_2 \leq (150N^4\eta^4 L^3 + 5L\eta^2 N^2)\|\nabla f(\boldsymbol{w}_t)\|^2 + 25\eta^4 L^3 N^3(\sigma^2 + 18N\Phi_t)$$

$$+ \frac{2\eta^2 N^2 L}{K} \sum_k dist(\boldsymbol{m}_{k,t}, \boldsymbol{m}_k^*)B^2 + 9\eta^2 N^2 L\Phi_t + \frac{NL\eta^2\sigma^2}{S}. \tag{45}$$

**Plugging $T_2$ and $T_1$ into R.H.S of Inequality (36)**, we obtain that:

$$\mathbb{E}\left[f\left(\boldsymbol{w}_{t+1}\mid\boldsymbol{w}_t\right)\right]\leq f\left(\boldsymbol{w}_t\right)-\eta N(\frac{1}{2}-150N^3\eta^3L^3-5L\eta N-15N^2\eta^2L^2)||\nabla f\left(\boldsymbol{w}_t\right)||^2$$

$$+(25\eta^4L^3N^3+\frac{5N^2\eta^3L^2}{2})(\sigma^2+18N\Phi_t)+\frac{4\eta^2N^2L+\eta N}{2K}\sum_k dist(\boldsymbol{m}_{k,t},\boldsymbol{m}_k^*)B^2$$

$$+9\eta^2N^2L\Phi_t+\frac{NL\eta^2\sigma^2}{S}.$$

$$(46)$$

Taking expectation over the randomness before round $t$ towards both sides of the inequality, it yields:

$$\mathbb{E}\left[f\left(\boldsymbol{w}_{t+1}\right)\right]\leq\mathbb{E}[f\left(\boldsymbol{w}_t\right)]-\eta N\kappa\mathbb{E}[||\nabla f\left(\boldsymbol{w}_t\right)||^2]+\rho_t. \tag{47}$$

where $\kappa=\frac{1}{2}-150N^3\eta^3L^3-15N^2\eta^2L^2-5N\eta L$ and $\rho_t=(25N^3\eta^4L^3+\frac{5N^2\eta^3L^2}{2})(\sigma^2+18N\Phi_t)+\frac{4N^2\eta^2L+N\eta}{2K}\sum_k dist(\boldsymbol{m}_{k,t},\boldsymbol{m}_k^*)B^2+9N^2\eta^2L\Phi_t+\frac{N\eta^2L\sigma^2}{S}$.

By rearranging, and summing from $t=0,\ldots,T-1$, the following result immediately shows:

$$\frac{1}{T}\sum_{t=0}^{T-1}\mathbb{E}[||\nabla f\left(\boldsymbol{w}_t\right)||^2\leq\frac{\mathbb{E}\left[f\left(\boldsymbol{w}_0\right)\right]-\mathbb{E}\left[f\left(\boldsymbol{w}_T\right)\right]}{T\eta N\kappa}+\frac{1}{T}\sum_{t=0}^{T-1}\rho_t$$

$$\leq\frac{f\left(\boldsymbol{w}_0\right)-f\left(\boldsymbol{w}^*\right)}{T\eta N\kappa}+\frac{1}{T}\sum_{t=0}^{T-1}\rho_t$$

$$(48)$$

Now we bound the local gradient over the personalized sparse model:

$$\frac{1}{TK}\sum_{t=0}^{T-1}\sum_{k=1}^{K}\mathbb{E}[||\nabla F_k\left(\tilde{\boldsymbol{w}}_{k,t}\right)||^2$$

$$=\frac{1}{TK}\sum_{t=0}^{T-1}\sum_{k=1}^{K}\mathbb{E}[||\nabla F_k\left(\tilde{\boldsymbol{w}}_{k,t}\right)-\nabla f(\boldsymbol{w}_t)+\nabla f(\boldsymbol{w}_t)||^2]$$

$$\overset{\text{Lemma 1}}{\leq}\frac{3}{TK}\sum_{t=0}^{T-1}\sum_{k=1}^{K}\mathbb{E}\left[||\nabla F_k(\tilde{\boldsymbol{w}}_{k,t})-\frac{1}{K}\sum_{k'}\boldsymbol{m}_k^*\odot\nabla F_{k'}(\tilde{\boldsymbol{w}}_{k',t})||^2\right.$$

$$\left.+||\frac{1}{K}\sum_{k'}(\boldsymbol{m}_k^*\odot\nabla F_{k'}(\tilde{\boldsymbol{w}}_{k',t})-\boldsymbol{m}_{k'}^*\odot\nabla F_{k'}(\tilde{\boldsymbol{w}}_{k',t}))||^2+||\nabla f(\boldsymbol{w}_t)||^2\right]$$

$$\overset{(a)}{\leq}\frac{3}{TK}\sum_{t=0}^{T-1}\sum_{k=1}^{K}((d-\beta)G^2+\beta B^2+\frac{1}{K}\sum_{k'}dist(\boldsymbol{m}_k^*,\boldsymbol{m}_{k'}^*)B^2)+\frac{3}{TK}\sum_{t=0}^{T-1}\sum_{k=1}^{K}\mathbb{E}||\nabla f(\boldsymbol{w}_t)||^2$$

$$\leq3(d-\beta)G^2+3\beta B^2+\frac{3}{K^2}\sum_k\sum_{k'}dist(\boldsymbol{m}_k^*,\boldsymbol{m}_{k'}^*)B^2+\frac{3}{TK}\sum_{t=0}^{T-1}\sum_{k=1}^{K}\mathbb{E}||\nabla f(\boldsymbol{w}_t)||^2$$

$$(49)$$

In inequality (a), we use the second claim in Lemma 8 to treat the first term, and the second term is bounded using a similar technique as in the first claim in Lemma 8.

Let $\Upsilon=3(d-\beta)G^2+3\beta B^2+\frac{3}{K^2}\sum_k\sum_{k'}dist(\boldsymbol{m}_k^*,\boldsymbol{m}_{k'}^*)B^2$ and plugging inequality (48) into the above inequality, it further implies that:

$$\frac{1}{TK}\sum_{t=0}^{T-1}\sum_{k=1}^{K}\mathbb{E}[||\nabla F_k\left(\tilde{\boldsymbol{w}}_{k,t}\right)||^2\leq\frac{3(f\left(\boldsymbol{w}_0\right)-f\left(\boldsymbol{w}^*\right))}{T\eta N\kappa}+\frac{3}{T}\sum_{t=0}^{T-1}\rho_t+\Upsilon \tag{50}$$

This shows the claim.

# D    DISCUSSION ON VARIOUS BOUNDS IN EXISTING STUDY

Under the non-convex setting, Theorem 1 bounds the squared gradient norm of the personalized iterates over the local loss. Existing studies on PFL zero in various different kinds of convergence bound, and under different assumptions. Below are some concrete examples:

- Li et al. (2021) study the upper bound of $\mathbb{E}\left[\|\tilde{\boldsymbol{w}}_{k,T} - \boldsymbol{w}_k^*\|^2\right]$ for any device $k$ (see their Corollary 1), where $\boldsymbol{w}_{k,T}$ is the personalized weights at iteration $T$ and $\boldsymbol{w}_k^*$ is the optimal weights under the client $k$'s local loss. Under the assumption of strongly convex, smoothness, bounded gradient, and gradient dissimilarity, as well as an additional assumption on the distance between personalized models and the optimal global model, it is theoretically proved that the proposed solution, named Ditto, achieves $O(1/T)$ convergence rate. Their derived convergence rate is on the same scale as the convergence rate of the global model produced by FedAvg. However, it is important to note that an extra assumption on the distance between personalized models and the optimal global model is required to achieve this rate.

- Hanzely et al. (2021) study the convergence of the proposed regularization problem (see their Theorem 4.5). Specifically, they propose to bound $\mathbb{E}\left[\sum_{k=1}^K \|\tilde{\boldsymbol{w}}_{k,T} - \boldsymbol{w}_k^*(\lambda)\|^2\right]$ where $\tilde{\boldsymbol{w}}_{k,T}$ is the personalized model, and $\boldsymbol{w}_k^*(\lambda)$ is the optimal model for their defined regularization problem under the regularized hyper-parameter $\lambda$. Under the strongly convex and smoothness assumption, the authors derive the bound as $\mathbb{E}\left[\sum_{k=1}^K \|\tilde{\boldsymbol{w}}_{k,T} - \boldsymbol{w}_k^*(\lambda)\|^2\right] \leq \left(1 - \frac{\alpha\mu}{n}\right)^T \sum_{k=1}^K \|\tilde{\boldsymbol{w}}_{k,T} - \boldsymbol{w}_k^*(\lambda)\|^2 + \frac{2n\alpha\sigma^2}{\mu}$. Further, by applying the variance reduction technique on their proposed L2GD solution, they eliminate the constant term (i.e., the second term) in the above bound. However, their convergence analysis are made towards a special case of the regularized problem. It remains unspecified about how and whether the personalized iteration produced by L2GD can achieve convergence towards the optimum for each client's local loss (i.e., the optimum of the ultimate PFL problem (P2)). In other words, the gap between $\tilde{\boldsymbol{w}}_{k,T}$ and $\boldsymbol{w}_k^* = \min_{\boldsymbol{w}_k} \hat{F}_k(\boldsymbol{w}_k)$ remains unknown.

- Under the non-convex setting, Deng et al. (2020) study the upper bound of squared gradient norm, or formally, $\frac{1}{T}\sum_{t=1}^T \mathbb{E}\left[\|\nabla F_k(\tilde{\boldsymbol{w}}_{k,t})\|^2\right]$ (see their Theorem 6) , where $\tilde{\boldsymbol{w}}_{k,t}$ is the personalized model for client $k$ and $\nabla F_k(\cdot)$ is the gradient of a model over the local loss. Under the smoothness and bounded variance conditions, they derive a bound with rate $O(\frac{1}{\sqrt{T}})$ to convergence. Similar to our results, a non-vanished residual presents in their bound. This non-vanished residual is related to: i) the gradient dissimilarity between clients' local loss function; and (ii) the gradient discrepancy, The form of this non-vanished term is highly similar to our derived residual term (i.e., $\Upsilon$ in our Theorem 5), since both the gradient dissimilarity factor (i.e., $G$ in Assumption 1) and gradient bound (i.e., $B$ in Assumption 4) would present in the non-vanished residual. Additionally, when the personalized factor is completely eliminated i.e., $\alpha_k \to 0$ in their formulation or $\beta \to d$ in our formulation, the non-vanished residual (i.e., $\Gamma$ in their formulation and $\Upsilon$ in ours) would both become significant to dominate the bound.

