# OpenReview forum: "On Heterogeneously Distributed Data, Sparsity Matters"
_ICLR.cc/2022/Conference — ICLR 2022 Submitted_

### Official Review · Reviewer_ECdR · 2021-10-25

**Correctness:** 2
**Technical Novelty And Significance:** 3
**Empirical Novelty And Significance:** 2
**Recommendation:** 5
**Confidence:** 4

**Main Review:**

My main comments are :

* The level or rigour needs to be improved. As a case in point, already the title suggests that the notion of "sparsity"
is crucial for the proposed approach. However, i was not able to find a rigorous definition of the quantity "sparsity".

* In my opinion the novelty of the proposed approach is limited. First of all, the sparse FPL problem is a special
case of the regularized FPL problem (3) using a specific choice for the regulariser (requiring the non-zero entries
of the i-th weight to agree for all clients. The regularised FPL has been studied in a sparse model context recently e.g. in

* M.Yamada, T. Koh, T. Iwata, J. Shawe-Taylor and S. Kaski, "Localized Lasso for High-Dimensional Regression", Proc. AISTATS 2017
* A. Jung, “Networked Exponential Families For Big Data over Networks,” in IEEE Access, Oct. 2020. doi: 10.1109/ACCESS.2020.3033817.
* Y. Sarcheshmehpour, M. Leinonen, A. Jung, “Federated Learning from Big Data over Networks,” to be presented at IEEE Int. Conf. on Acoustics, Speech and Sig. Proc., 2021.

it might be useful to compare your approach with the (generalised linear model based) methods proposed in above papers.

* The FedSpa algorithm seems to be a straightforward combination of existing techniques (stochastic gradient descent and masking search techniques
for sparse deep net training).

* The relevance and usefulness of Theorem 1 is unclear. How did you use Theorem 1 for applying FedSpa (e.g., choice of hyper-parameters)
in the numerical experiments of Section 5? Im also not sure if (5) can be referred to as "error bound" as it actually
bounds the gradient of the objective function along the trajectory of the algorithm. However, the numerical experiments do not
use this gradient as the figure of merit but the resulting accuracy. How is (5) relevant for bounding the accuracy of FedSpa?

* The clarity or presentation and use of language needs significant improvement:

- what are "fully dense PFL models" ?
- what is "consistently cheap communication"?
- "The Non-IID data distribution makes the training process prohibitively difficult, and the obtained global model may not fit all the local data."
try to be more specific
- ".. is another challenging problem that remains mysterious."
- "FedSpa does not deploy a single global model, but allows each client to own its unique sparse model..." this is a bit confusing as FedSpa actually tightly couples
the sparse model parameters of each client. as i understood, they must have the same non-zero weight value for each individual feature.
- what is a "evolutionary sparse model" ?
- "... which is agnostic to all kinds of network architectures ..." with is meant by "being agnostic" here ? i believe that at least the assumptions on the loss
function somehow interact with the particular network architecture.

- "Experimental results ... also coincides with the theoretical conclusion." Pls explain more precisely what you mean by "theoretical conclusion".

- "We emphasize that three main progresses are made towards the interpretable and SOTA compression-based PFL..." interpretability is a hot topic in machine learning.
However, it is not clear how your approach ensures interpretability. Why is FedSpa interpretable?

- The masks appearing in (P4) need more discussion. Are these fixed but unknown (ground truth), or just some estimate.

- "Mimicking the data-parallel solution ..." which solution are you referring here exactly?

- "The framework of FedSpa is extensible ..."

- would it be possible to avoid "dist(m1,m2)" by using the ell0 norm (which is already used elsewhere).

- what is "cosine annealing " ?

- pls explicitly state any input/hyperparamter used in Algorithm 1,2 3 E.g. Algorithm 3 might have a random seed as its input.

- "Our main focus of analysis is on the second and third non-vanished terms while communication round..."

- " The result in Eq. (5) is highly related to setting of sparse volume,..." ?

- " ... to perform update." which update ?


**Summary Of The Paper:**

This paper proposes an interesting approach to FL with heterogeneous client data. The main idea is
to learn a common weight vector for all clients but allow them to individually mask this global weight vector.

****
I appreciate the effort that authors put into adressing my concerns which have been partially covered. Thus, i have slighltly raised the grade of my recommendation. However, i would ask the area chair to verify if the authors have addressed all my concerns satisfactorily. In particular, im still not sure about the relation between (P4) and regularized PFL (P3) which is quite generic and also allows for reguarlizers that are indicator functions of sets.



**Summary Of The Review:**

My main comments are :

* The level or rigour needs to be improved. As a case in point, already the title suggests that the notion of "sparsity"
is crucial for the proposed approach. However, i was not able to find a rigorous definition of the quantity "sparsity".

* In my opinion the novelty of the proposed approach is limited. First of all, the sparse FPL problem is a special
case of the regularized FPL problem (3) using a specific choice for the regulariser (requiring the non-zero entries
of the i-th weight to agree for all clients. The regularised FPL has been studied in a sparse model context recently e.g. in

* M.Yamada, T. Koh, T. Iwata, J. Shawe-Taylor and S. Kaski, "Localized Lasso for High-Dimensional Regression", Proc. AISTATS 2017
* A. Jung, “Networked Exponential Families For Big Data over Networks,” in IEEE Access, Oct. 2020. doi: 10.1109/ACCESS.2020.3033817.
* Y. Sarcheshmehpour, M. Leinonen, A. Jung, “Federated Learning from Big Data over Networks,” to be presented at IEEE Int. Conf. on Acoustics, Speech and Sig. Proc., 2021.


it might be useful to compare your approach with the (generalised linear model based) methods proposed in above papers.

* The FedSpa algorithm seems to be a straightforward combination of existing techniques (stochastic gradient descent and masking search techniques
for sparse deep net training).

* The relevance and usefulness of Theorem 1 is unclear. How did you use Theorem 1 for applying FedSpa (e.g., choice of hyper-parameters)
in the numerical experiments of Section 5? Im also not sure if (5) can be referred to as "error bound" as it actually
bounds the gradient of the objective function along the trajectory of the algorithm. However, the numerical experiments do not
use this gradient as the figure of merit but the resulting accuracy. How is (5) relevant for bounding the accuracy of FedSpa?

---

> ### Author Response · Authors · 2021-11-21
> **Authors Response (3/3)**
>
> > "..what is meant by being agnostic here? I believe that at least the assumptions on the loss function somehow interact with the particular network architecture."
>
> **Response.** We state that our method is agnostic to model architecture because we do not need to manually adjust our method (either on its hyper-parameter or its inner logic) if the network architectures being used are different. We now refrain from using the word "agnostic" in this submission, but state that the problem "can be applied to most of the model architectures without model-specific hyper-parameter tuning".
>
> > " Experimental results ... also coincide with the theoretical conclusion." Pls explain more precisely what you mean by "theoretical conclusion".*
>
> **Response.** Please see our revision on this sentence: "Experimental results demonstrate the superiority of FedSpa and also coincide with the theoretical conclusion -- with the rise of data heterogeneity, setting a higher sparsity of FedSpa may potentially result in a better convergence on its personalized models."
>
> > "We emphasize that three main progresses are made towards the interpretable and SOTA compression-based PFL..." interpretability is a hot topic in machine learning. However, it is not clear how your approach ensures interpretability. Why is FedSpa interpretable?
>
> **Response.** Before making a solution interpretable, we first have to formally formulate the problem we aim to solve. So to achieve good interpretability, we need to establish a clear problem statement for compression-based PFL . Before this study, there is no clear problem formulation available for compression-based PFL (see previous works (Li et al., 2020) (Vahidian et al., 2021)). Besides, the model update process of compression-based PFL is not rigorously established in a formal form in both two studies. Therefore, we state that our work (with rigorous problem statement and update rule) is a solid step towards interpretable sparse model-based PFL.
>
> > The masks appearing in (P4) need more discussion. Are these fixed but unknown (ground truth), or just some estimate.
>
> **Response.** Yes. The masks $m_k^*$ are fixed but unknown.
>
> > "Mimicking the data-parallel solution ..." which solution are you referring to here exactly?
>
> **Response:** We refer to the data-parallel SGD solution here. This solution is the vanilla solution for distributed machine learning, and is a special case of FedAvg (in the setting of full participation and one single local step).
>
>  > "The framework of FedSpa is extensible ..."
>
> **Response.** We state "The framework of FedSpa is extensible "  because one can explore diversified mask searching techniques and plug arbitrarily one of them into our FedSpa framework. We make the following description to enable easier grasp. "Specifically, we use the mask surrogate $m_{k,t}$ in Eq. (3) to perform update, which allows us to plug in arbitrary mask searching techniques to determine the iterate process of $m_{k,t}$."
>
> > Would it be possible to avoid "dist(m1,m2)" by using the ell0 norm (which is already used elsewhere).
>
> **Response.** Thanks for the good advice! It is indeed possible to avoid "dist(m1,m2)" by using the ell0 norm. We can think of one possible way, i.e., $dist(m1,m2)=|| m1 \oplus m2 ||_0$. We are not sure if this formulation would be more intuitive to our readers. We will be happy to hear further advice from the reviewer.
>
> > what is "cosine annealing " ?
>
> **Response.** "Cosine annealing" is an essential sub-procedure for DST. We perform cosine annealing to decay the pruning rate each round, such that the mask exploration would be radical in the initial stage while weakening in the later stage.   In this submission, we give a detailed explanation of the cosine annealing sub-procedure. Please refer to Appendix A for details.
>
> > pls explicitly state any input/hyperparameter used in Algorithm 1,2 3 E.g. Algorithm 3 might have a random seed as its input.
>
> **Response:** Thanks for pointing out the presentation issue! Indeed, our algorithm 3 needs a random seed as input. We have clearly specified the input of the proposed algorithms in this submission.
>
> > "Our main focus of analysis is on the second and third non-vanished terms while communication round..."
>
> **Response.** We make the following revision to the mentioned statement: "We focus our analysis on the second and third residuals, which do not vanish as communication round $T \to \infty$.
>
> > "The result in Eq. (5) is highly related to the setting of sparse volume,..."?
>
> **Response.** "The result in Eq. (5) is highly related to the setting of sparse volume,..." -> The sparse volume, i.e., $\beta$ could have a drastic impact on the upper bound shown in Eq. (5).
>
> >" ... to perform update." which update?
>
> **Response.** " ... to perform update." -> "to perform update on the global model $w_t$ (see Eq. (3))".

---

> ### Author Response · Authors · 2021-11-21
> **Authors Response (2/3)**
>
> > The relevance and usefulness of Theorem 1 is unclear. How did you use Theorem 1 for applying FedSpa (e.g., choice of hyper-parameters) in the numerical experiments of Section 5?
>
> **Response:** Thanks. We give Theorem 1 mainly in an attempt to support our conclusion that making the personalized model sparse (to a proper extent) could lead to a better convergence result of the ultimate problem (P2) . Our experimental results also confirm this conclusion. Please see Figure 1, Figure 4 and Table 3. We show that the sparsity that leads to higher accuracy is around 0.4 in a Non-IID setting. However, it is indeed not straightforward to use the results in Theorem 1 to determine the optimal sparsity. We now refrain from exaggerating the usefulness of Theorem 1 in the new submission.
>
> >  I am also not sure if (5) can be referred to as "error bound" as it actually bounds the gradient of the objective function along the trajectory of the algorithm. However, the numerical experiments do not use this gradient as the figure of merit but the resulting accuracy.
>
> **Response:** Thanks for pointing out the presentation issue. Our goal is actually to let the personalized models converge to the local optimum of Problem (P2), or in other words,  to let the gradient of personalized models converge 0  (given that the gradient of local optimum equals to 0). So, perhaps it would be better to name this bound as "Convergence bound of personalized models". We have made this revision in the new submission.   In our experiments, we indeed use accuracy instead of gradient norm as the main metric to perform the evaluation. Our main motivation to do so is that making the accuracy higher is the ultimate objective of a machine learning task. And we do believe the following relationship between gradient, loss, and accuracy is clear to normal readers: smaller gradient norm <->  smaller loss <-> higher classification accuracy, so we did not perform extra experiments over metrics other than accuracy, like gradient norm.
>
> > How is (5) relevant for bounding the accuracy of FedSpa?
>
> **Response:** Thanks. It is indeed more intuitive to derive the bound over the accuracy of FedSpa. But we are afraid that bounding the accuracy of FedSpa is not easy to achieve. Existing research on federated learning either bound the distance (or $l_2$ norm ) between the optimal weights and the iterated weights, or bound the squared gradient norm (just like ours), but we cannot trace any existing studies that propose to directly bound on accuracy.  However, having bound on accuracy could be a solid direction to work on in the future.  From the comments, it seems that normal readers could experience difficulty in understanding the meaning of our bound. So, in this submission, we summarize various bounds already presented in the existing literature of PFL to enable an easier grasp. Please see Appendix D for detail.
>
>
> > what are "fully dense PFL models" ?
>
> **Response.** Thanks. We make this revision to enable better understanding:  "fully dense PFL models" -> "an intact (or dense) PFL model"
>
> > what is "consistently cheap communication"?
>
> **Response.**Thanks. The following revision is made to improve readability. Consistently cheap communication" -> "cheap communication, computation, and memory cost.
>
> > The Non-IID data distribution makes the training process prohibitively difficult, and the obtained global model may not fit all the local data." try to be more specific
>
> **Response.** We rewrite the sentence: "Global model produced by weight average (or FedAvg and its non-personalized variants) exhibit unsatisfactory performance in a Non-IID data distribution setting."
>
> >".. is another challenging problem that remains mysterious."
>
> **Response.** ".. is another challenging problem that remains mysterious." -> ".. is another challenging problem that remains unresolved."
>
> > "FedSpa does not deploy a single global model, but allows each client to own its unique sparse model..." this is a bit confusing as FedSpa actually tightly couples the sparse model parameters of each client. as i understood, they must have the same non-zero weight value for each individual feature."
>
> **Response.** There may be some misunderstanding here. FedSpa tightly couples the dense (but not sparse) parameters of each client.   We state that each client owns its unique sparse model because each client has different masks (i.e., m_k^*), indicating that they have different weight coordinates that are set to zero. Clearly, exactly due to the listed reason, the parameters are not totally the same for all clients, and therefore we say that they have a unique sparse model.
>
> > What is an evolutionary sparse model?
>
> **Response.** The model training can be seen as an evolutionary (or iterative) process. We allow the local training of clients to be always based on a sparse model, even though they are different from each other in different iterations (so the models are evolutionary and sparse).

---

> ### Author Response · Authors · 2021-11-21
> **Thanks for the valuable feedback! Below are our responses. (1/3)**
>
> Thanks for the valuable feedback.  We address the raised issues as follows:
>
> >The level of rigor needs to be improved. As a case in point, already the title suggests that the notion of "sparsity" is crucial for the proposed approach. However, I was not able to find a rigorous definition of the quantity "sparsity".
>
> **Response.** Thanks. We acknowledge that the definition of “sparsity” is not rigorously established in our previous submission, as we thought that it is a quite standard notion for experts in related disciplines. In this submission, we have footnoted its meaning in its first appearance.
>
> > In my opinion the novelty of the proposed approach is limited. First of all, the sparse FPL problem is a special case of the regularized FPL problem (3) using a specific choice for the regulariser (requiring the non-zero entries of the i-th weight to agree for all clients. The regularised FPL has been studied in a sparse model context recently.
>
> **Response.** Thanks for the invaluable comment. However, there may be a misunderstanding over our proposed SPFL problem, i.e., problem (P4). In fact, *problem (P4) in its current form is not a special case of the regularization problem*, because it cannot be identically transformed to one.  To address your concern, we try to formulate a similar problem using a regularizer. One *possible regularization problem* could be:
>
> \begin{equation}
>   \begin{split}
> (P5) \min_{\{ w , w_{1}, \cdots, w_{K} \}}  \frac{1}{K} \sum_{k=1}^{K}  \mathbb{E}[ L_{( x,  y) \sim D_k } ( w_k; ( x,  y)) ] +\lambda \sum_{k} ||    w_{k} - m_k^* \odot w ||_1 ,
>   \end{split}
> \end{equation}
>
> But *problem (P5) is not equivalent to our sparse personalized problem (P4)*. To justify this statement, we transform problem (P4) to its *equivalent problem (P6)*:
> \begin{equation}
>   \begin{split}
> (P6) & \min_{ w,\{ w_{1}, \cdots, w_{K} \}}  \frac{1}{K} \sum_{k=1}^{K}  \mathbb{E}[ L_{( x,  y) \sim D_k } (w_k  ; ( x,  y)) ] \\\\
> s.t.   & \quad  w_k = m_k^* \odot w \quad \text{ for any } k&
>   \end{split}
> \end{equation}
>
> Clearly, *problem (P6) is not equivalent to the regularization problem (P5)* because: (i) there is a hard constraint in (P6), which cannot be directly transformed into the regularization term, and clearly, the optimal solution of (P5) may not necessarily comply this hard constraint in (P6); (ii) there is an extra hyper-parameter $\lambda$ in (P5), which does not appear in (P6). Therefore, we can conclude that problem (P5) is not equivalent to (P6), and consequently, the regularization problem (P5) is not equivalent to (P4). Other forms of regularization problems cannot be equivalent to (P4) based on the same two reasons. So, we insist that problem (P4) is not a special case of the regularization problem.
>
> > it might be useful to compare your approach with the methods proposed in the above papers.
>
> **Response.** Thanks. Those are very good references. We would like to incorporate the reference solutions in our baselines. However, we encounter some difficulties in their implementation.  Firstly, as we stated before, there is no evidence that our optimization problem is a special case of the regularization problem, and therefore the regularization-based methods in the provided reference cannot be directly applicable to our original problem (P4). Secondly, even though we view the regularization problem (P5) as the problem we like to optimize, we experience some difficulties in applying the recommended solutions to this particular problem, as in the following: (i) the reference solutions are all based on linear-model, so cannot be directly applied to a deep neural network in our regime; (ii), even making the loss function in the problem (P5) to an MSE loss with the linear model, the provided solutions cannot deal with the extra constant term $m_k^*$ in problem (P5).  Due to the encountered challenges, we are not able to include the reference solutions in the comparison. However, we do incorporate a discussion on the provided reference in Section 2. Please see our revision marked in blue color.
>
> > The FedSpa algorithm seems to be a straightforward combination of existing techniques (stochastic gradient descent and masking search techniques for sparse deep net training).
>
> **Response.** Thanks. We indeed borrow the idea from Dynamic Sparse Training (DST) to search for the optimal masks. But we deny that our solution is a straightforward combination of SGD and DST.  Please see our update rule in Eq. (3) and Eq. (4). Our solution contains a mask ($m_{k,t}$) in its update process, which certainly does not present in general SGD. We guess the misinterpretation from this reviewer stems from the bolded name "FL-adapted SGD", so in the new submission, we rename it to FL-adapted Update. Also, as we state in Section 4.2, we have modified the DST technique to do mask searching in the FL context. Therefore, we deny that our solution is a  direct combination of DST and SGD.

---

> ### Author Response · Authors · 2021-11-25
> **Thanks for your response. Below are our further comments about the relation between (P4) and regularized PFL (P3).**
>
> > I appreciate the effort that authors put into addressing my concerns which have been partially covered. Thus, i have slightly raised the grade of my recommendation. However, i would ask the area chair to verify if the authors have addressed all my concerns satisfactorily. In particular, im still not sure about the relation between (P4) and regularized PFL (P3) which is quite generic and also allows for regularizers that are indicator functions of sets.
>
> **Response.** Indeed,  regularized PFL (P3) is quite generic, which allows us to replace it with arbitrary regularizers. And you bet,  the regularizers can be indicator functions of sets. However, though the two problems share some commonality,  we still insist that  (P4) is *not a special case of regularized problem in which the penalty parameters $\lambda$ can be set to arbitrary values*. To demonstrate this,  assume we have three clients (i.e., K=3). Assume client 1 only maintains one piece of data: $(x_{1}=(1,2), y_{1}= 1) $ in its local data distribution; client 2 maintains this piece of data $(x_{2}=(2,1), y_{2}= 2)$; and client 3 maintains this piece of data $(x_{3}=(2,2), y_{3}= 3)$.   We like to use linear model with 2 dimensions to solve the task, i.e.,  $w, w_k \in \mathbb{R}^2$. Suppose the optimal masks for the three clients are respectively $m_1^* = (1,0)$, $m_2^* = (0,1)$ and $m_3^* = (1,0)$ . Then our problem (P4) in this case can be:
> \begin{equation}
>   \begin{split}
> (P4) \qquad & \min_{\{ w, w_1, w_2, w_3\}}  \frac{1}{3} \sum_{k=1}^{3}  || x_k^T w_k- y_k ||^2 \\\\
> s.t. \qquad & w_k = m_k^* \odot w  \quad \text{for } k=1,2,3
>   \end{split}
> \end{equation}
> As per the above setting, it is straightforward to verify that the optimal solution of the above problem are:
>
> | $w_1^*$  | $w_2^*$  | $w_3^*$  |  $w^*$ |
> | :-----------: |:-----------:|:-------------:|-------------:|
> |(1.4, 0)|(0, 2)|(1.4, 0)|(1.4, 2)
>
> Now we study one particular regularized problem with a penalty function, which appears to be the most similar one to our problem (P4), as shown in the following:
>
> \begin{equation}
>   \begin{split}
> (P5) \quad \min_{w, w_1, w_2, w_3 }  \frac{1}{3} \sum_{k=1}^{3}  || x_k^T w_k- y_k ||^2 + \frac{1}{3}\sum_{k=1}^{3} \lambda ||w_k - m_k^* \odot w ||^2.
>   \end{split}
> \end{equation}
>
> We note that the optimal solutions (i.e., $w_k$ and $w$) of this regularized problem depends on the exact choices on penalty parameter $\lambda$. We just list some specific choices of $\lambda$ and the corresponding optimal solutions below:
>
> |  $\lambda$ | $w_1^*$  | $w_2^*$  | $w_3^*$  |  $w^*$ |
> | ------------- | :-----------: |:-----------:|:-------------:|-------------:|
> |0.1|(1.023,0.0)|(0.0,1.996)|(1.49,0.0)|(1.258,1.961)
> |0.5|(1.094,0.0)|(0.0,1.996)|(1.48,0.0)|(1.283,1.991)
> |1|(1.151,0.0)|(0.0,1.998)|(1.46,0.0)|(1.305,1.997)
> |5|(1.301,0.0)|(0.0,1.999)|(1.42,0.0)|(1.362,1.999)
> |10|(1.343,0.0)|(0.0,2.0)|(1.41,0.0)|(1.378,2.0)
> |20|(1.369,0.0)|(0.0,2.0)|(1.41,0.0)|(1.388,2.0)
>
> As shown, the optimal solution of the regularized problem is different under the different settings of penalty parameter $\lambda$. The optimal solution of (P5) can only be the same with our (P4) iff $\lambda \to \infty$. So, if you insist, our problem indeed  can be written into a regularized problem if the $\lambda$ is confined to an infinite positive value. But the regularized objective is not an exact penalty function (whose solutions should be the exact solution to the original problem (P4) under a finite value of  $\lambda$). In other words, this means you have to solve an infinite sequence of regularized problems for different $\lambda$ to obtain the same solution of (P4).  Consequently, we argue that transforming our problem (P4) into such a form would not be so technically meaningful.
>
> Thanks again for your response! We would be happy to hear further advice from you and the area chair!

---

> ### Author Response · Authors · 2021-11-28
> **[last day reminder] As the deadline is approaching, would you mind checking if our clarification makes sense to you?**
>
> Dear Reviewer ECdR,
>
> We appreciate your comments and time! We have made further clarification on the relation between SPFL problem (P4) and regularized PFL problem (P3).  Would you mind checking it and confirming if you have further questions about it? Thanks in advance!
>
> Best regards,
>
> Authors

---

### Official Review · Reviewer_Zg2F · 2021-10-29

**Correctness:** 3
**Technical Novelty And Significance:** 2
**Empirical Novelty And Significance:** 2
**Recommendation:** 5
**Confidence:** 3

**Main Review:**

**1.** The idea of using sparse masks to model personalization for federated learning is not novel in this work. Prior works utilize this idea with other techniques (Li et al., 2020) (Vahidian et al., 2021). Moreover, several side-benefits such as low communication cost, cheaper computation, and fewer memory requirements should also be attributed to those original works where sparse masks are used, and the same side-benefits of sparsity were mentioned.

**2.** As far as I can see, the idea of current work that differs from the above two mentioned papers is how the sparse masks are handled throughout the training process. However, there does not seem to be done much progress in this direction neither. The authors nominate two heuristics on how to evolve sparse masks over the iterates, one of which is simply fixing the same mask for all clients and iterations. However, isn’t it the same as initially training the smaller model for all clients?

**3.** It was mentioned in the contributions that the problem of sparse PFL is rigorously formulated (I guess this refers to the last part of section 3, Sparse PFL problem (P4)). However, you do not provide a deeper understating of this formulation, e.g., How well it models personalization compared to other formulations? Why it makes sense to use a small subset of the global model (i.e., sparse masks) as the local, personalized model beyond what was heuristically proposed by (Li et al., 2020) (Vahidian et al., 2021)? Can the formulation (P4) be regarded as a particular case of regularized PFL (P3), say with $l_1$ norm?

**4.** The algorithm FedSpa seems to directly apply FedAvg (LocalSGD with partial participation) to the problem (P4). What is the novelty in the algorithm itself?

**5.** Theoretical analysis is very weak. Let us ignore the first two terms in the rate (5) and consider the third non-vanishing term $\Upsilon$. Using only Assumption 4 and running any method, one can upper bound the left-hand side of (5) by $d B^2$.

**6.** The title is somehow vague and does not describe the work well. The paper is on personalized federated learning, which is not reflected in the title.


**Summary Of The Paper:**

This work studies personalized federated learning through sparse local masks. For each client, the local model is subset of the global model. The paper proposes and analysis a new method, named FedSpa, which trains sparse local models for all clients reducing communication size, the amount of computation and memory cost.

**Summary Of The Review:**

The level of originality is low given the prior works. A new formulation lacks deeper understanding and theoretical analysis is weak.

---

> ### Author Response · Authors · 2021-11-21
> **Authors Response (3/3)**
>
> > The algorithm FedSpa seems to directly apply FedAvg (LocalSGD with partial participation) to the problem (P4). What is the novelty in the algorithm itself?
>
> **Response.** We indeed mimic FedAvg to solve Problem (P4). But we note that FedSpa has a significant difference with FedAvg. Before we justify the novelty of FedSpa, we first summarize its training workflow in one communication round, as follows:
>
> (i) Server produces and distributes the local models $\tilde w_{k,t}$ based on the personalized mask of client, i.e., $\tilde w_{k,t,0} = m_{k,t} \odot w$.
>
> (ii) Clients perform  a mask-involved SGD update (but not vanilla SGD) on their sparse models, i.e.,
> $$\tilde w_{k, t, \tau+1}  = \tilde w_{k,t, \tau}- \eta m_{k,t} \odot \nabla_{\tilde w_{k,t,\tau} } L( \tilde  w_{k,t,\tau}; \xi_{k,t,\tau}), \tau=0, 1, \dots, N-1,$$
> (iii) clients do mask searching to find the optimal masks.
> (iv) clients upload the sparse updates $\tilde w_{k,t,0} - \tilde w_{k,t,N}$, which are then averaged by the server. Finally, the averaged update is applied into the global model in previous round i.e.,
> $$ w_{t+1} = w_{t} - \frac{1}{ | S_t| } \sum_{k \in S_t}   (\tilde w_{k,t,0} - \tilde w_{k,t,N})$$
>
> Then we highlight the following novelty of FedSpa. Firstly, FedAvg distributes the same global model to all clients, while FedSpa produces and distributes local models in step (i). Secondly,  FedAvg utilizes vanilla SGD in the local update process, while FedSpa utilizes a mask-involved SGD update in step (ii). Thirdly, FedSpa includes an extra mask searching step (iii), which does not exist in FedAvg.
> Finally, with FedAvg, clients update the model weights, which are then averaged by the server. On contrary, in step (iv) of FedSpa, clients upload the sparse gradient update, which is subsequently averaged by the server. The averaged update is applied to the global model to conclude a round of training.
>
> > Theoretical analysis is very weak. Let us ignore the first two terms in the rate (5) and consider the third non-vanishing term $\gamma$. Using only Assumption 4 and running any method, one can upper bound the left-hand side of (5) by $d B^2$.
>
> **Response.** Thanks for the comment. Yes, it is true that one can always upper bound the left-hand side of Eq. (5) by $d B^2$ where $d$ is the dimensions of weights. However, we note that $d$ is a pretty large value (could be in the scale of millions for a deep neural network), so directly applying assumption 4 may not promise us a meaningful bound.  Another observation is that the non-vanished residual $\gamma$ would indeed turn to the scale of  $d B^2$ when the sparse volume $\beta \to d$. In this case, no trainable parameters could exist in the model (i.e., all the parameters are 0), so the bound in this case is in the same scale of direct utilization of Assumption 4. We present our theoretical analysis mainly in an attempt to interpret our empirical observation that properly sparsifying the model could ensure a better convergence result, i.e., the gradient norm of the sparse personalized models could have a lower bound, compared to the solution that trained on an intact network.
>
> >The title is somehow vague and does not describe the work well. The paper is on personalized federated learning, which is not reflected in the title.
>
> **Response.** Thanks for pointing out the issue regarding the title of this paper. We in this submission have re-named the title as "Achieving Personalized Federated Learning with Sparse Local Models".

---

> > ### Comment · Reviewer_Zg2F · 2021-12-08
> > **Thanks to the authors for the response and my apologies for the late reply.**
> >
> > **Clear optimization problem:** Yes, I can see that you have formulated an optimization problem for sparse-based PFL. However, the novelty in it is *very small* as you did not do it from scratch, but rather mathematically formalized what is written in (Li et al., 2020) (Vahidian et al., 2021) in words: for example, LTN (Lottery Ticket Network) in (Li et al., 2020) is nothing else than the masked version of the global weights, which is explicitly mentioned in their Algorithm 1 for instance. Next, you claim to clarify the ultimate problem of FL with (P2). However, as you said there is no consensus on what should the objective function be in PFL, so (P2) cannot be the ultimate problem. Thus, the gap on the actual ultimate objective for PFL is still there.
> >
> > **Different and formalized model update rule.** My reasoning is similar to the above point. You formalized mathematically what was informally mentioned in prior works using words. For example, in (Li et al., 2020) it is written *"we propose to seek the LTN of each client during each communication round, and then communicate only the parameters of LTNs between the clients and the server in FL. After aggregating the LTNs of the clients, the server will distribute the updated parameters of the corresponding LTN to each client. Finally, a personalized model, instead of a shared global model, will be learned at each client. Since the LTN is determined by pruning the base model using the local data of each client, the data-dependent features have already been incorporated in the LTN."*
> >
> > **Different mask searching techniques and numerical observations.** I agree that the paper has merits in proposing mask searching techniques and providing numerical evidence to support. This is the main reason that keeps my score around borderline and not lower. It is strange to see RSM as a separate mask searching technique since personalization is completely ignored, as you mentioned in this case.
> >
> > **Theoretical Analysis.** Let me reemphasize how weak the current theory (rate (5) to be precise) is to me. So, the $\Upsilon$ term contains $(d-\beta)G^2 + \beta B^2$ plus some other term, which could be as bad as $O(d B^2)$. Suppose we assume for a moment that $G^2=B^2$, then $\Upsilon \ge d B^2$. However, if we change the infinity norm in Assumption 4 with worst-case bound, then for ***any method***, the right-hand side of (5) can be replaced by smaller (or at least not bigger) quantity $d B^2$. Of course, $G^2$ could be smaller than $B^2$ and maybe choosing $\beta\sim d$ is better, but ***it is not clear what the strength of the rate (5) is compared to the assumption it uses.***
> >
> > In short, the paper provides improved numerical evidence that local masks are useful in PFL. I agree that there is ***a little improvement*** in this direction. However, the core idea of using masks for personalization is already presented in prior works, and it does not bring a new idea to the field. In terms of theoretical contributions, the novelty is ***very small***. Some informal statements from prior works are made more formal through mathematical formulations, which seem pretty straightforward. These formulations are used to develop theory, which is very weak, as discussed above. ***To conclude, I lean towards rejection and would like to keep my score.***

---

> > > ### Author Response · Authors · 2021-12-09
> > > **Thanks for the critical comments. Still, the authors would like to make the following clarification.**
> > >
> > > Thanks for the critical comments from this reviewer. Still, the authors would like to make the following clarification on the mentioned issues:
> > >
> > > **Insufficient novelty on the SPFL problem and no consensus on the ultimate PFL problem:** Firstly, we admit that our SPFL problem is somehow similar to the idea described in the previous two papers. However, we claim the formulation to be one of our contributions not because of its technical difficulty, but because of its paramount importance in subsequent research on sparse PFL problems. One cannot do subsequent research on SPFL without knowing what's the real objective, which to our best knowledge, has not been formalized until this submission. Secondly, there is indeed no consensus on the ultimate PFL problem, but this does not mean we cannot build one. We are actually calling for a consensus that the proposed problem (P2) is what PFL is solving. Solving other transformed problems (like the regularized problem, or our SPFL problem) is in essence to approximate the solution of (P2).
> > >
> > > **Insufficient novelty on update rule:** We need to emphasize that *the update rule of our solution and that of the previous work are not the same*. Our solution proposes to average over the sparse gradient update from clients. On contrary,  LotteryFL (Li et al., 2020) proposes to "communicate only the parameters of LTNs between the clients" and the server "aggregate the LTNs of the clients". To illustrate their difference in update rule of global model, we give the following example. Say, the global model in the beginning of a communication round is $w_g=(1,1,2)$ (with 3 dimensions), and the sparse global weights distributed two clients are respectively $w_1=(0 ,1,2)$ and $w_2= (1,0,2)$. Then the two clients perform local training, and produce the following gradient update: $g_1=(0,-2,-3)$ and $g_2=(-3,0,-2)$ . Correspondingly, the sparse parameters of local models (i.e., the LTNs) would be $\bar{w}_1 = w_1- g_1=(0,3,5)$, and $\bar{w}_2 = w_2-g_2=(4,0,4)$. Then the average results (i.e., updated global model) of LotteryFL would be  $w_g^{ltn} = \frac{1}{2}(\bar{w}_1+\bar{w}_2) = (2,\frac{3}{2},\frac{9}{2})$. On contrary, our proposed solution FedSpa would produce the next global model as: $w_g^{spa} = w_g-\frac{1}{2}(g_1+g_2) = (\frac{5}{2},2,\frac{9}{2})$. As shown, since $w_g^{spa} \neq w_g^{ltn}$, the generated global models are different between FedSpa and LotteryFL.
> > >
> > > **Strange to see RSM as a separate mask searching technique:** Indeed, by adopting RSM, personalization is completely eliminated. But we still like to involve it in our mask searching technique because we would like to provide more diversified observations on how the mask searching technique would affect the overall performance of FedSpa framework.
> > >
> > > **Weak theoretical analysis:** Thanks! We would consider your advice, and look over our proof to figure out if there is a possibility to make the upper bound tighter.

---

> ### Author Response · Authors · 2021-11-21
> **Authors Response (2/3)**
>
> >The authors nominate two heuristics on how to evolve sparse masks over the iterates, one of which is simply fixing the same mask for all clients and iterations. However, isn’t it the same as initially training the smaller model for all clients?
>
> **Response.** Indeed, our RSM solution is essentially fixing the same mask for all clients and iterations, and it is exactly the same with using FedAvg to train on a smaller model. The reason we propose RSM is that we like to explore how the mask setting would affect the training performance in different data settings. For RSM, all the masks are the same, so personalization is completely eliminated.  Our experiment shows that RSM achieves satisfactory performance in IID setting, but obtains inferior accuracy in Non-IID setting. This seems to tell that maintaining different masks (or to achieve personalization) indeed benefits the training performance in Non-IID setting, but would possibly degrade the performance in the IID setting.  We propose the RSM solution mainly with the aim to show this conclusion.
>
> >It was mentioned in the contributions that the problem of sparse PFL is rigorously formulated (I guess this refers to the last part of section 3, Sparse PFL problem (P4)). However, you do not provide a deeper understating of this formulation, e.g., How well it models personalization compared to other formulations? Why it makes sense to use a small subset of the global model (i.e., sparse masks) as the local, personalized model beyond what was heuristically proposed by (Li et al., 2020) (Vahidian et al., 2021)?
>
> **Response.** Thanks for the comments. The sparse PFL problem proposed in (P4) is indeed one of the contributions of this paper. The key problem of PFL is how to build the connection (i.e., to federate) among the client's local models, while maintaining some degrees of personalization in the local models. However, we are afraid that there is not a fair and universal way to quantitively analyze how well a problem itself can model personalization. We would like to leave this for future work.
>
> Here is our interpretation of why the sparse personalized models perform better in the Non-IID setting. We know that weights in a model serve some particular functions, e.g., extracting some specific features, forming representation, etc. Original FL aims to develop a global model that can make accurate classification on all classes of data. For example, in CIFAR10 we have 10 labels of data. The global model trained by general FL needs to make accurate classification on data of all labels, and therefore all functions (or equivalently, all weights) in the model are essential.  However, do the clients really need all the functions (or weights) to perform their own tasks? Maybe not necessarily true in the Non-IID setting. For example, if the clients only host data with some specific labels, and its task is to make classification on data with these same labels. Then maybe some functions e.g., extracting particular features from data that it never has,  may not be necessary, or could even inversely compromise the performance of the client's own task.  Consequently, it is better to drop the weights that perform these functions and form the client's own sparse local models (i.e., to be personalized).
>
> >Can the formulation (P4) be regarded as a particular case of regularized PFL (P3), say with $l_1$ norm?
>
> **Response:** We insist that problem (P4) cannot be regarded as a  particular case of regularized PFL (P3), since it cannot be identically transformed to one. We first try to adapt our problem (P4) into a regularization problem with $l_1$ norm, as follows:
>
> \begin{equation}
>   \begin{split}
> (P5) \min_{\{w ,w_{1}, \cdots, w_{K} \}}  \frac{1}{K} \sum_{k=1}^{K}  \mathbb{E}[ L_{( x,  y) \sim D_k } ( w_k; (x,  y)) ] +\lambda \sum_{k=1}^K ||   w_{k} -\boldsymbol m_k^* \odot w ||_1 ,
>   \end{split}
> \end{equation}
>
> But *problem (P5) is not equivalent to our sparse personalized problem (P4)*. To prove this statement, we transform problem (P4) to its *equivalent problem (P6)*:
> \begin{equation}
>   \begin{split}
> (P6) & \min_{w,\{  w_{1}, \cdots, w_{K} \}}  \frac{1}{K} \sum_{k=1}^{K}  \mathbb{E}[ L_{(x,  y) \sim D_k } (w_k ; (x,  y)) ] \\\\
> s.t.   & \quad  w_k = m_k^* \odot w \quad \text{ for any } k&
>   \end{split}
> \end{equation}
>
> Clearly, *problem (P6) is not equivalent to the regularization problem (P5)* because: (i) there is a hard constraint in (P6), which cannot be directly transformed into the regularization term.  (ii) there is an extra hyper-parameter $\lambda$ in (P5), which does not appear in (P6).  Then we can conclude that problem (P5) is not equivalent to (P6), and consequently, the regularization problem (P5) is not equivalent to (P4). Other forms of regularization problems cannot be equivalent to (P4) based on the same reasons. As such, we justify that problem (P4) is not a special case of the regularization problem.

---

> ### Author Response · Authors · 2021-11-21
> **We thank the reviewer for the invaluable comments. Below are our responses (1/3).**
>
> We thank the reviewer for the invaluable comments. Below are our responses to the raised issues:
> > The idea of using sparse masks to model personalization for federated learning is not novel in this work. Prior works utilize this idea with other techniques (Li et al., 2020) (Vahidian et al., 2021). Moreover, several side-benefits such as low communication cost, cheaper computation, and fewer memory requirements should also be attributed to those original works where sparse masks are used, and the same side-benefits of sparsity were mentioned.
>
> **Response.** You bet, FedSpa is a follow-up work that improves upon existing methods. However, we need to highlight several merits that make FedSpa outstands the existing methods:
> * **Clear optimization problem**. We formulate a clear optimization problem for FedSpa.  This is the first time the optimization problem of sparse-models-based PFL is rigorously formulated. Besides, in the problem formulation part, we clarify the ultimate problem of personalized federated learning to be Problem (P2). Though there are various problem formulations of PFL problem (e.g., those problems with regularizer), there is still not a common consensus on what is exactly we like to optimize in PFL. Inspired by the current gap, we make a clear statement over the ultimate optimization problem of PFL in this paper.
> * **Different and formalized model update rule.** We clearly show how the sub-models are aggregated in the server. While prior works either propose to ”aggregating the Lottery Ticket Network via FedAvg” (Li et al., 2020), or ”taking the average on the intersection of unpruned parameters in the network” (Vahidian et al., 2021), we instead formulate the aggregation as averaging the sparse updates from clients. Our aggregation process is substantially different from the previous works since clients only transmit the sparse update, but not the sparse models, to the server for aggregation.  Besides, we use rigorous mathematical formulation to formalize our update rule (see Eq. (3) and (4)), but the previous works merely use informal languages to describe their update process.
> *  **Different mask searching techniques.** The two prominent previous works adopt iterative pruning to achieve model compression in FL. That is to say, the training process starts from an intact model, which is then iteratively pruned to the target sparsity.  On contrary, we adopt sparse-to-sparse training in FedSpa. That is to say, the local training is based on a sparse model from the beginning of the training process, and the masks continue to evolve (towards the optimal masks) in the training process.  The edge of sparse-to-sparse training in FL is outstanding, since: (i) clients (typically mobile devices) may not have enough memory to store the intact model, so training from an intact model is simply unrealistic. (ii) the communication overhead is reduced since the iterative pruning solution still has to exchange the dense (or nearly dense)  local models in the early phase of training.
> * **Better results and new empirical observations.** As per our evaluation of three datasets, FedSpa has significant performance improvement over Sub-FedAvg (Vahidian et al., 2021). Also, we derive a new observation that adopting local models with different sparse masks may lead to degradation of training performance in the IID setting, which has not been reported elsewhere. Exactly due to this observation, we further propose and evaluate another sparse-to-sparse solution named FedSpa (RSM), which essentially is to make all the masks static. Surprisingly, this solution remains almost the same performance as the vanilla FedAvg in the IID setting.
>
> * **Theoretical Analysis.** We conduct theoretical analysis to show how the sparsity of models would affect the convergence towards the local optimum of the ultimate problem (P2).
>
> >  As far as I can see, the idea of current work that differs from the above two mentioned papers is how the sparse masks are handled throughout the training process. However, there does not seem to be done much progress in this direction neither.
>
> **Response.** Thanks. The major difference between our works and the above-mentioned papers is indeed the sparse-to-sparse mask searching process. We propose two masking searching strategies, namely, Dynamic Sparse Training (DST) and Random and Static Maks (RSM). For DST, we actually have made some degrees of modification to adapt it into the FL setting.  Firstly, the pruning is performed individually by each client based on their local models, and the gradient used for weights recovery is derived using the client’s local training data. Secondly, once the next masks are generated, we propose to warm-start the training process via inheriting the regrown weights from the global model. Please refer to the remark below Algorithm 2 for our modification over DST.

---

> ### Author Response · Authors · 2021-11-28
> **[last day reminder]  Would you mind checking our response？**
>
> Dear Reviewer Zg2F,
>
> Thanks for the valuable comments! We have tried our best to address your concerns. Here is a summary of our detailed response below. We humbly expect you can check it and confirm whether our response has addressed your concerns.
>
> * We have highlighted our novelty that outstands from the previous works (Li et al., 2020) (Vahidian et al., 2021).
>
> * We have made a clarification on our real motivation to propose the simple yet effective RSM solution, which essentially is to fix the random masks throughout the training.
>
> * We provided an interpretation on why the local models, which inherits a subset of parameters from the global model, could work in the case that local data is heterogeneously distributed.
>
> * We made a clarification that our proposed SPFL problem (P4) is not a special case of the regularized PFL (P3), with detailed reasons listed in our response.
>
> * We explained why the left-hand side of Eq. (5) can always be upper bounded by $dB^2$. We also added more discussions on the theoretical results from the existing study. The bound we derive seems to share some commonalities with that in  APFL (Deng et al., 2020). Please see our discussion in Appendix D for details.
>
> * The title has been modified to "Achieving Personalized Federated Learning with Sparse Local Models".
>
> Thanks a lot for your feedback! We would sincerely appreciate it if you could check our response.
>
> Best Regards,
>
> Authors

---

### Official Review · Reviewer_PGV4 · 2021-10-31

**Correctness:** 3
**Technical Novelty And Significance:** 2
**Empirical Novelty And Significance:** 2
**Recommendation:** 6
**Confidence:** 3

**Main Review:**

Strengths: Addressing the data heterogeneity and reducing communication cost are an important problems in federated learning. Using one architecture for all local models and different sparse masks seems a new idea in the field.

Weaknesses: 1) At every iteration, the Algorithm 1 FedSpa requires new masks to be computed, which highly increases the computational time on local machines. The experiments results only report the communication time without the total running time. It is not clear whether the wall clock time can be reduced or the contrary. Also the FedSpa requires communication every iteration which causes more communication cost.

2) One merit of using sparse masking is that it reduces the communication memory size and the Table 1 measures the communication cost in terms of GB, so a natural baseline would be FedAvg with compression techniques. A lack of such comparison makes the true effect of sparse masking questionable.

**Summary Of The Paper:**

This submission proposes federated learning with personalized sparse mask (FedSpa) to apply sparse masks to the local models in training and communication, hence to make the model adaptive to its local data and also to reduce communication cost.


**Summary Of The Review:**

This paper has proposed a new method to address the heterogeneously distributed data problem in federated learning. However, the algorithms has some impractical components and the effects over state-of-the-art is not convincing enough due to lack of some baselines.

---

> ### Author Response · Authors · 2021-11-21
> **Many thanks for the valuable feedback.**
>
> Many thanks for the valuable feedback. We would like to address your concern as below:
> >  At every iteration, the Algorithm 1 FedSpa requires new masks to be computed, which highly increases the computational time on local machines. The experiment results only report the communication time without the total running time. It is not clear whether the wall clock time can be reduced or the contrary.
>
> **Response:** Thanks. FedSpa indeed needs extra computation to search for the optimal masks, but this does not account for significant time for the local computation, because the mask searching is only performed after multiple epochs of local training (say, 5 epochs). To confirm our previous statement, we do an extra experiment to show the wall time respectively for local training and mask searching procedures. The table below shows the results.
>
> |  Task | Wall Time  (Train)  | Wall Time  (Mask Search)  | Ratio (Mask Search/Train) |
> | ------------- | :-----------: |:-----------:|:-------------:|
> |EMNIST-LeNet (CPU) | 1.03$\pm$0.04s | 0.09$\pm$0.0s | 8.92\%$\pm$0.6|
> |CIFAR10-VGG11 (CPU) | 11.4$\pm$0.25s | 2.19$\pm$0.11s | 19.19\%$\pm$1.09|
> |CIFAR100-Resnet18 (CPU) | 28.61$\pm$0.38s | 3.7$\pm$0.28s | 12.93\%$\pm$1.08|
> |EMNIST-LeNet5 (GPU) | 0.39$\pm$0.01s | 0.02$\pm$0.0s | 5.66\%$\pm$0.25|
> |CIFAR10-VGG11 (GPU) | 1.56$\pm$0.03s | 0.22$\pm$0.01s | 14.3\%$\pm$0.48|
> |CIFAR100-Resnet18 (GPU) | 2.71$\pm$0.01s | 0.33$\pm$0.01s | 12.06\%$\pm$0.2|
>
> As shown, the mask searching process only accounts for a small portion of wall time (approximately 5 \%-20 \%) of the entire computation time on the local devices.  These experimental results have also been appended to our ablation study in Appendix B.4 in the new submission.
>
> >Also the FedSpa requires communication every iteration which causes more communication costs.
>
> **Response:** Thanks. FedSpa does not need to communicate after every SGD update. Similar to Local SGD (or FedAvg), we only require the devices to be synchronized (or perform communication) after several epochs of training. Additionally, since FedSpa only transmits the sparse update to perform synchronization, the communication cost of FedSpa should be reduced compared with FedAvg.
>
> > One merit of using sparse masking is that it reduces the communication memory size and Table 1 measures the communication cost in terms of GB, so a natural baseline would be FedAvg with compression techniques. A lack of such comparison makes the true effect of sparse masking questionable.
>
> **Response:** Thanks for the good advice. We in this submission further incorporate Subsampling, a gradient compression technique originally proposed in [1],  into our evaluation. Please refer to our revision in Table 1, Table 2 and Figure 3. As shown, Subsampling exhibits significant performance degradation (specifically, slower convergence rate and lower final accuracy), compared with the uncompressed solution (e.g., FedAvg). On contrary, our proposed solution could achieve even better accuracy and faster convergence, compared with the uncompressed solution FedAvg.
>
> [1] Konečný J, McMahan H B, Yu F X, et al. Federated learning: Strategies for improving communication efficiency[J]. arXiv preprint arXiv:1610.05492, 2016.

---

> > ### Comment · Reviewer_PGV4 · 2021-11-24
> > **Some concerns addressed.**
> >
> > Thank you for the new experiment reports and clarifying. Overall I raise my score to 6. Some of my concerns are addressed. However, the computation time for the masking seems non-trivial and would increase the complexity in practice.

---

> > > ### Author Response · Authors · 2021-11-24
> > > **Re-concern of computation time for mask searching**
> > >
> > > Thanks for raising the recommendation score to 6! As per our experimental results, the computation time for masking indeed accounts for non-negligible extra time (about 5% to 20%) for local computation. But we would like to note the following points to see if they can erase your practical concern:
> > >
> > > * One of our proposed strategies named FedSpa-RSM would not need to perform this mask searching process, and therefore would not need extra computation. Specifically, FedSpa-RSM is essentially to make each client initialize in the first round a random mask, which remains unchanged in the later round of training. In Figure 6, we demonstrate that FedSpa-RSM with different initialization achieves about 62% accuracy in CIFAR100 Non-IID setting B, which outperforms the non-personalized solution FedAvg in the same setting (which is about 45.0% accuracy, see Table 1).  So, if the extra time for computation is really the problem,  it is okay to employ FedSpa-RSM with different initialization (rather than FedSpa-DST) in the Non-IID FL training, though as per Figure 6, FedSpa-RSM with different initialization indeed suffers a certain amount of performance loss in accuracy than FedSpa-DST (which requires extra time for mask searching).
> > >
> > > * Proportion of time used for mask searching can be reduced if we set the local epochs to be larger. In our experiment, we set the local epochs to 5. In other words, the clients would perform mask searching after 5 epochs of local training. If the local epochs are set to a higher value, say 10 epochs, then the time proportion of mask searching time could be approximately reduced by half.
> > >
> > > * Barring the training time, time used for communication is also slowing down FL. One extra benefit of our proposed solution is that it can shorten the time used for communication in both the downlink and uplink process because the size of the model being synchronized in each round is reduced. So, conceivably, the time used for mask searching would become less significant if we calculate its proportion in one whole round of FL training (that is: the sum of downlink+uplink communication time and local training time). To demonstrate this, under different bandwidth constraints, we are planning to measure the whole time in one single round of FL training (including communication and training). We would leave further comments to you once the experimental results come out, if this could help address your concern.
> > >
> > > *  Finally, the time used for mask searching can be optimized with specific system-level designs. Our preliminary testing results over time for mask searching are based on the vanilla PyTorch implementation. However, further system-level optimization can be done towards our mask searching process. For example, one can hide the mask searching in the local training process. Say, assume a client needs to do 5 epochs of local training before uploading the models to the server for averaging. Here if we allow this client to perform mask searching after 4 epochs of training, while parallelly performing the 5-th epoch of the local training process, then the extra time for mask searching can be hidden into the local training process. As FedSpa is still a preliminary study, we would like to leave this system-level optimization for future work, but one thing for sure is that the mask searching time has a large space to optimize with better system-level designs.

---

> > > ### Author Response · Authors · 2021-11-26
> > > **Experimental results of communication and computation time tested on RaspBerry Pi.**
> > >
> > > To erase the practical concern from this reviewer. We have deployed FL training on a Raspberry Pi 4 Model B [1] with 8GB RAM.  The task we tested is EMNIST, the model we used is LeNet5, and each round of training involves 5 epochs of training over 125 pieces of data. In this experiment, under different bandwidth limitations, we measured the training time and communication time in one round of FL training, and our results are shown in the following table:
> > >
> > > |  Bandwidth Setting | Communciation Time (C)  |  Training Time (T) | Ratio (T/(T+C)) |
> > > |:------------- :| :-----------: |:-----------:|:-------------:|
> > > |u: 2 Mbps d: 5 Mbps | 7.29 s$\pm$0.97 | 62.57 s$\pm$0.23 | 11.66\%$\pm$1.55 |
> > > |u: 3 Mbps d: 6 Mbps | 7.75 s$\pm$0.99 | 45.61 s$\pm$0.17 | 16.99\%$\pm$2.2 |
> > > |u: 4 Mbps  d: 7 Mbps| 7.84 s$\pm$0.38 | 36.24 s$\pm$0.03 | 21.64\%$\pm$1.06 |
> > > |u: 5 Mbps d: 8 Mbps| 8.71 s$\pm$0.56 | 30.48 s $\pm$0.19 | 28.58\%$\pm$1.85 |
> > > |u: 6 Mbps d: 9 Mbps | 8.63 s$\pm$0.82 | 27.33 s$\pm$0.23 | 31.57\%$\pm$2.71 |
> > >
> > > *u means the maximum uplink bandwidth, and d means the maximum downlink bandwidth.
> > >
> > > Our results seem to demonstrate that communication time accounts for a critical amount of time in one single round of training (approximately 68%- 88% in our testbed). This means the extra computation time, which typically accounts for an extra 5%-20% of the training time would not be so significant.  Further, this overhead seems to be minor if considering the benefit (i.e., reduction of communication, memory, computation, and improvement in training accuracy) brought by the mask searching.
> > >
> > > Thanks. The authors are looking forwards to further discussion on the practical issue.
> > >
> > > [1] Raspberry Pi. https://www.raspberrypi.com/products/raspberry-pi-4-model-b/

---

> > > ### Author Response · Authors · 2021-11-28
> > > **[last day reminder]  Would you mind updating the scores and checking our response？**
> > >
> > > Dear reviewer PGV4,
> > >
> > > Thanks for the effort and time put into reviewing this paper. As the deadline for this round of review is approaching, would you mind having an update on the recommendation score (as suggested in the previous comment)?
> > >
> > > We have tried to address your practical concern over the extra computation time. Here is a summary of our responses:
> > >
> > > *  One of our FedSpa-RSM would not need extra computation time for mask searching.
> > >
> > > *  Proportion of time used for mask searching could potentially be reduced in a different setting of hyper-parameters.
> > >
> > > * Communication time accounts for a large part of elapsed time in one round of training, which has been verified by our new experiments on real IoT devices. Therefore the extra mask searching time, which accounts for about 5%-20% of the local training time would not be so significant within one round of training.
> > >
> > > * Time used for mask searching can be optimized with specific system-level designs.
> > >
> > > We hope our responses could help address your concern on the practical aspects of our proposed solutions. Once again, we thank the reviewer for the constructive feedback.
> > >
> > > Best regards,
> > >
> > > Authors

---

### Official Review · Reviewer_mQ15 · 2021-11-02

**Correctness:** 3
**Technical Novelty And Significance:** 3
**Empirical Novelty And Significance:** 3
**Recommendation:** 6
**Confidence:** 3

**Main Review:**

Strengths:

1: This paper proposes a novel personalized federated learning scheme by incorporating sparse-to-sparse searching techniques.

2: Extensive experiments are conducted to verify the superiority of the proposed algorithm.

3: This paper is well-written and easy to follow. The background knowledge is presented well. The readers can easily understand the paper.

Weaknesses:

1: The theoretical analysis in Theorem 1 is unclear and weak. It is unclear that what the error bound in Theorem 1 means. The authors need to analyze and compare the theoretical results to other comparable methods.

2: The title is ambiguous and may lead to inappropriate reviewers.

3: I see no code attached to this submission, which makes me a bit concerned about reproducibility.



**Summary Of The Paper:**

This paper studies personalized sparse training for federated learning. The proposed method FedSpa is a novel personalized federated learning scheme that employs personalized sparse masks to customize sparse local models on the edge.  The authors provide the theoretical result with regard to the error bound and empirical results with several personalized federated learning methods.

**Summary Of The Review:**

I marginally accept this paper mainly due to its novelty and better experimental results.

After reading the response, I keep my score at 6.

---

> ### Author Response · Authors · 2021-11-21
> **Thanks for your approval and the helpful comments.**
>
> We sincerely appreciate the supportive and invaluable comments from the reviewer. Below are our response to your concerns.
>
> > The theoretical analysis in Theorem 1 is unclear and weak. It is unclear that what the error bound in Theorem 1 means.
>
> **Response.** Thanks for pointing out the issue in our theoretical analysis. Indeed,  the term "error bound" may be a rather vague term to express the meaning of our bound. Our goal is actually to let the personalized models converge to the local optimum of Problem (P2), or in other words,  to let the gradient of the obtained personalized models converge 0  (given that the gradient of local optimum equals to 0). The bound specifies how the squared gradient norm of the converged personalized models would have deviated from 0.  So, perhaps it would be better to name this bound as "Convergence bound of personalized models". We have made this revision in the new submission.
>
>
>
> > The authors need to analyze and compare the theoretical results to other comparable methods.
>
> **Response.** Thanks for the good suggestion. We now append a discussion on the connection between our result and the convergence result of Ditto[1],  L2GD[2], and APFL[3]. Please see Appendix D for our discussion.
>
> > The title is ambiguous and may lead to inappropriate reviewers.
>
> **Response.** Thanks. We agree with this reviewer that the title of our initial submission is somehow vague and might potentially lead to confusion. For sake of clearness, we re-name the title to "Achieving Personalized Federated Learning with Sparse Local Models".
>
> > I see no code attached to this submission, which makes me a bit concerned about reproducibility.
>
> **Response.** Thanks. We in this submission include our code in the supplementary material.
>
> [1] Li T, Hu S, Beirami A, et al. Ditto: Fair and robust federated learning through personalization[C]//International Conference on Machine Learning. PMLR, 2021: 6357-6368.
>
> [2] Hanzely F, Richtárik P. Federated learning of a mixture of global and local models[J]. arXiv preprint arXiv:2002.05516, 2020.
>
> [3] Deng Y, Kamani M M, Mahdavi M. Adaptive personalized federated learning[J]. arXiv preprint arXiv:2003.13461, 2020.

---

### Author Response · Authors · 2021-11-30
**Final summary from Authors**

Dear Area Chair and Reviewers,

We appreciate your efforts in reviewing our paper and providing constructive comments to help us revise the paper.

After going through a careful revision in the rebuttal process,  it is a pleasure for us to see that the merits of our work have been recognized by reviewers mQ15 and PGV4. Thank you!

We also like to thank reviewer ECdR for raising the recommendation score as recognition of addressing most of the critical concerns. However, it seems that reviewer ECdR still remains uncertain about the relationship between our formulated SPFL problem (P4) and the regularized problem (P3). As suggested in our response to this issue, we insist that **the PFL problem (P4) is not a special case of regularized problem (P3)**. To defend our statement, we provide a simple yet instructive example to do the analysis. We demonstrate by this example that a specific problem with a penalty term in its objective can be identical to our problem (P4) iff the penalty hyper-parameter $\lambda \to \infty$. However, typically for a regularized problem, the control parameter $\lambda$  can be set to arbitrary values, but not being constrained to $\lambda \to \infty$.    Humbly, as suggested by reviewer ECdR, we would like to ask the area chair to join our discussion on this specific concern, if it is possible.

We appreciate the careful and detailed review from the reviewer Zg2F, but we are yet to hear a further response from him/her. We genuinely hope in having a possible discussion with reviewer Zg2F to figure out if we have resolved the given concerns.   From the initial review comments, we guess that the main concern from this reviewer stems from the novelty aspect. (That is, how our work differs from previous works [1] [2]).  In response, we summarize our main novelty into the following aspects:
*  We formulate a clear optimization problem for FedSpa, while the previous work did not present a similar problem definition.
*  We apply different and formalized model update rules. Specifically, we propose to average over the sparse gradient in the server, rather than the sub-model as suggested in the previous works.
*  We employ different mask searching techniques. Specifically, FedSpa adopts a sparse-to-sparse training paradigm, which means the model trained in the local training phase is sparse from the beginning of the training process. This is in sharp contrast to the previous works that propose to iteratively prune an intact local model towards a sparse personalized model.
*  Our solution achieves better performance (in terms of accuracy, convergence speed, and communication complexity). Additionally, we derive a new empirical observation that adopting local models with different sparse masks may lead to degradation of training performance in the IID setting.
*  We conduct theoretical analysis to show how the sparsity of models would affect the convergence towards the local optimum of the ultimate problem (P2), which is not available in the previous study.

We really hope that our submission could provide a new perspective on the compression-based PFL (or sparse PFL), allowing others in the FL community to build new ideas upon it. Once again, we sincerely thank the area chair and reviewers for the time and effort put into reviewing this submission!


Best regards,

Authors

[1] Li A, Sun J, Wang B, et al. Lotteryfl: Personalized and communication-efficient federated learning with lottery ticket hypothesis on non-iid datasets[J]. arXiv preprint arXiv:2008.03371, 2020.

[2] Vahidian S, Morafah M, Lin B. Personalized Federated Learning by Structured and Unstructured Pruning under Data Heterogeneity[J]. arXiv preprint arXiv:2105.00562, 2021.

---

### Decision · Program_Chairs · 2022-01-20

**Decision:**

Reject

**Comment:**

Dear authors,

I have carefully read the reviews, rebuttals and the subsequent discussion. The review scores are mixed (5, 5, 6, 6). Let me comment on some of the key issues raised by the reviewers. I will elaborate on some of them with my own insights.

1) You insist that (P4) cannot be regarded as a particular case of (P3). But this is trivially incorrect. The hard constraint in the reformulation (P6) you mention in the discussion can be written as a regularizer: *indicator function* of the constraint set. Indeed, let  $\cal C$ be the set of points $W=(w_1,\dots,w_K)$ for which there exists $w$ such that $w_k = m_k^* \circ w$ for all $k$. Then the regularizer defined by ${\cal R}(W) = 0$ if $W\in {\cal C}$ and ${\cal R}(W) = +\infty$ if otherwise does the job. This is a well defined regularizer. Such regularizers are routinely used in optimization to model hard constraints. So, the formulation you consider is a special case of (P3).  Moreover, as pointed out by Reviewer Zg2F, and acknowledged by the authors, "The idea of using sparse masks to model personalization for federated learning is not novel in this work. Prior works utilize this idea with other techniques (Li et al., 2020) (Vahidian et al., 2021). Moreover, several side-benefits such as low communication cost, cheaper computation, and fewer memory requirements should also be attributed to those original works where sparse masks are used, and the same side-benefits of sparsity were mentioned." The claim that one of the novel contributions of FedSpa is "we formulate a clear optimization problem for FedSpa" is weak, especially in the light of the above comment, and the "moral" existence of the formulation in prior work, albeit not expressed in a mathematical notation. The fact that previous works did not formulate this properly is a major issue with those works, and not a major contribution of this work. A clear mathematical formulation of what one wants to achieve should be a standard requirement. In any case, I appreciate the clarity nevertheless.

2) The same reviewer states that the key idea of the paper that differs from the above two mentioned papers is how the sparse masks are handled. One of the two ideas proposed is trivial and is equivalent to standard non-personalized FL (if all masks are the same, the submodes they defined can be considered a global model). The second idea does not seem to have any interesting/distinctive theoretical support.

3) Sparse-to-sparse training in FedSpa may be novel, but the claim that "the masks continue to evolve (towards the optimal masks) in the training process" is not supported by theory nor experiments. If indeed you can show that the local masks evolve to some meaningful notion of an optimal mask, this would be interesting.

4) I also agree with the other points raised by this reviewer. I have read the author response to these comments. (BTW: Language such as "you bet" is inappropriate). While some of them make sense, they do not reduce the severity of the concerns by a large enough margin.

5) The comment about the weakness of the main theorem is particularly concerning. Indeed, the main theorem may be vacuous, and the authors need to do a thorough explanation of the result and its importance (on its own and in comparison with existing literature and rates). I do not believe such a comparison could be advantageous to the proposed method though. The expressions are complicated. It seems that for any meaningful mask size, the non-vanishing term will be too large. The theorem is not a valid convergence result as the authors do not show that the right hand side can indeed be provably made arbitrarily small by some choice of the parameters of the method. For instance, it is not guaranteed that $dist(m_{k,t}, m_k^*)$ will converge to zero. In this sense, calling this theorem "Convergence of personalized models" is incorrect and misleading. This is a fatal issue, unfortunately. The authors should make it absolutely clear that the result does not prove convergence.

6) Assumptions 1, 2, and 4 are very strong. For example, Assumption 2 is not provably satisfied for lower bounded nonconvex smooth functions when subsampling (=minibatching) is used to produce the stochastic gradient. Assumption 3 is also quite strong: it is not satisfied by convex quadratics. Assumption 1 is also strong - most recent works on FL do not require any similarity assumptions.

In summary, while this direction of research is interesting, the level of contributions in this work is marginal at best. The key theoretical result is misleading in that it does not imply convergence while it is marketed as such. Moreover, strong assumptions (relative to what is achieved in the latest papers) are used to obtain it. Because of these concerns, and other concerns raised by the reviewers, I do not have any other choice but to reject the paper.

Area Chair